EMBO
Molecular Medicine

# Ki-67 promotes inflammatory signaling governing neutrophil recruitment during respiratory infections

Min Yee[1], Ravi Misra[1], Sarah Vesecky [1], Michael Barravecchia[1], Rauf A Najar[1], Arshad Rahman [1], Gloria S Pryhuber [1], David A Dean [1], B Paige Lawrence[2], Daniel Fisher [3] & Michael A O'Reilly [1✉]

## Abstract

**Neutrophils defend against respiratory infections but cause acute lung injury (ALI) when excessively recruited to the lung. Early life environmental factors can shape lung development, but how they impact neutrophil recruitment is not known. We show that exposing newborn mice to hyperoxia increases the number of adult alveolar type 1 (AT1) epithelial cells expressing the proliferation marker Ki-67. Although these cells were not proliferating, they expressed high levels of chemokines that stimulated neutrophil recruitment and ALI when mice were infected with influenza A virus or exposed to lipopolysaccharide (LPS). Neutrophil recruitment and chemokine production were attenuated in Ki-67 hypomorph mice infected with virus or exposed to LPS and enhanced by genetically overexpressing Ki-67 in their lungs. Silencing Ki-67 in a mouse AT1-like cell line reduced basal and IL-1β stimulation of RelA/p65 and NF-κB-dependent transcription of the chemokines *Cxcl1* and *Cxcl5*. Our findings reveal a novel role for Ki-67 to modulate the intensity of epithelial pro-inflammatory signaling, controlling neutrophil recruitment. The severity of respiratory infections may be influenced by mitogens and environmental factors that increase the expression of Ki-67.**

**Keywords** Acute Lung Injury; Alveolar Epithelial Cells; Influenza A Virus; Mice; Susceptibility
**Subject Categories** Evolution & Ecology; Immunology; Microbiology, Virology & Host Pathogen Interaction

## Introduction

Respiratory viruses and bacteria are a significant cause of morbidity and mortality throughout the world (Flerlage et al, 2021). Acute lung injury occurs when pathogens directly infect and cause an overwhelming loss of alveolar epithelial cells, vital for lung function. For example, avian influenza virus H5N1 is highly pathogenic because it infects and kills alveolar type (AT) 2

epithelial cells independently of inflammation (Salomon et al, 2007; van Riel et al, 2006). Since AT2 cells produce surfactant, express innate immune genes, and self-renew or differentiate into AT1 cells, their loss during infection compromises alveolar homeostasis and regeneration. An excessive inflammatory response during infection can also kill alveolar epithelial cells. Neutrophils serve as the first line of innate immune defense because they have a broad arsenal of anti-pathogen functions and are rapidly recruited to sites of infection by a well-orchestrated pattern of chemokine (C–X–C motif) ligands, otherwise known as CXCL chemokines (Chan et al, 2021; Rosales, 2020). However, too many neutrophils cause extensive alveolar injury and fatal lung disease, such as when humans succumb to influenza or SARS-CoV-2 infections (Flerlage et al, 2021). Similarly, increased neutrophil recruitment and excessive lung injury is seen in mice infected with the highly pathogenic mouse-adapted PR8 (H1N1) strain of influenza A virus (IAV) (Brandes et al, 2013; Tate et al, 2011). In contrast, acute lung injury is not seen when neutrophils are immune depleted during infection or when mice are infected with Hk31 (H3N2), a less pathogenic mouse-adapted strain of IAV that poorly stimulates neutrophil recruitment. Despite a deep understanding of how neutrophils function during infection, it is unclear why they are excessively or inappropriately recruited in some individuals.

Early gene–environment interactions that shape postnatal lung development may also govern how the lung responds to pathogens and inhaled pollutants (Virolainen et al, 2023). An inappropriate oxygen environment at birth is a classic example because it alters postnatal lung development and increases the severity of respiratory infections later in life. Infants and children born at high altitude develop lungs better suited to respire in hypoxia, but at the expense of being more susceptible to respiratory viral illness (Choudhuri et al, 2006). Conversely, supplemental oxygen (hyperoxia) given to preterm infants causes alveolar simplification and enhances the severity of respiratory infections (Townsi et al, 2018). This oxygen-driven susceptibility to respiratory illness following infection has been modeled in rodents. Exposing newborn mice to 75% oxygen for the first two weeks of life increases inflammation and airway hyperreactivity when they are infected with rhinovirus as adults (Cui et al, 2021). Similarly, exposing newborn mice to 100% oxygen for the first 96 h of life causes persistent inflammation and fibrosis when they are infected

[1]Division of Neonatology, Department of Pediatrics, School of Medicine & Dentistry, The University of Rochester, Rochester, NY, USA. [2]Department of Environmental Medicine, School of Medicine & Dentistry, The University of Rochester, Rochester, NY, USA. [3]Institut de Génétique Moléculaire de Montpellier, CNRS, INSERM, Université de Montpellier, Montpellier 34293, France. ✉E-mail: michael_oreilly@urmc.rochester.edu

as adults with Hkx31 (H3N2), which poorly recruits neutrophils and thus rarely causes disease in control mice exposed to room air (Dylag et al, 2021; O'Reilly et al, 2008). In this model, neonatal hyperoxia increases the severity of alveolar epithelial injury that cannot be efficiently repaired because hyperoxia also causes alveolar simplification and thus reduces the number of AT2, which are required to restore alveolar integrity (Yee et al, 2017). Neonatal hyperoxia does not change viral tropism or clearance (Domm et al, 2017; Giannandrea et al, 2012); thus, how it promotes alveolar epithelial injury during infection is not known.

While mapping how neonatal hyperoxia depletes AT2 cells in the adult lung, we made the unexpected and surprising finding that it increases the number of AT1 cells expressing Ki-67. Ki-67 has historically been used to identify proliferating cells because it is a nuclear protein that is highly expressed throughout the cell cycle but not in long-term quiescent cells (Sobecki et al, 2017). However, Ki-67 is not required for cell proliferation (Andres-Sanchez et al, 2022). For example, Ki-67 continues to be expressed by osteosarcoma cells even when proliferation is chemically or genetically inhibited (van Oijen et al, 1998). Deleting the Fzr1 gene encoding an activator of the E3 ubiquitin ligase responsible for Ki-67 degradation causes persistent expression of Ki-67 in differentiated tissues such as the cerebellum of mice (Sobecki et al, 2016). But this does not cause inappropriate cell proliferation. Genetic mutations that disrupt expression of Ki-67 in MCF-10A, DLD-1, NIH-3T3, HeLa, and other cell lines do not impair proliferation (Cidado et al, 2016; Cuylen et al, 2016; Mrouj et al, 2021; Sobecki et al, 2016). Similarly, TALEN-mediated disruption of Ki-67 in mice does not impair their growth, development, or tissue homeostasis (Sobecki et al, 2016). While this is arguably the strongest evidence that Ki-67 is not essential for proliferation, it raises the question of Ki-67's functions in the cell. A growing number of studies indicate Ki-67 plays an important role in shaping chromatin. For example, Ki-67 promotes formation of nucleolar material along the mitotic chromosome periphery (Booth et al, 2014; Sobecki et al, 2016), which prevents chromosome collapse and dispersion of DNA during mitosis (Cuylen et al, 2016). Ki-67 can act in concert with p53 to prevent mitotic DNA damage and activation of cell cycle checkpoints during DNA replication (Garwain et al, 2021). It can modulate the intensity of gene expression in various tumor cells by organizing the nuclear location of tri-methylated histone H3K9 and H4K20 (Sobecki et al, 2016). Ki-67 can also bind the nucleolar protein NIFK (also known as MKI67 interacting protein) (Takagi et al, 2001), which is a conserved protein required in yeast for rRNA processing (Oeffinger and Tollervey, 2003). Recently, Ki-67 was shown to play a critical role in all stages of carcinogenesis, most notably modulating the expression intensity of genes that render tumors visible to the immune system (Mrouj et al, 2021). Thus, Ki-67's ability to maintain DNA integrity and modulate the intensity of gene expression may help cells respond to their environment.

Here we show that Ki-67 expressed by AT1 cells enhances the intensity of pro-inflammatory signals that promote neutrophil recruitment and thus the severity of acute lung injury when mice are infected with influenza A virus or challenged with lipopolysaccharide (LPS). Neonatal hyperoxia increases the number of adult AT1 cells expressing Ki-67, thus driving excessive neutrophil recruitment and the severity of ALI caused by infection. Our findings add to a growing body of evidence that Ki-67 modulates the expression level of genes by which cells respond to their environment. The severity of respiratory infections may therefore be governed by mitogens and early environmental factors, such as neonatal hyperoxia, that increase Ki-67 expression.

# Results

## Neonatal hyperoxia increases expression of Ki-67 in non-proliferating adult AT1 cells

Newborn mice were exposed to room air or hyperoxia between postnatal days 0 and 4 (PND4) and then all mice were exposed to room air until they were 8 weeks old (Fig. 1A). While mapping how hyperoxia affects cell proliferation, we discovered that it increases the number of alveolar cells expressing the proliferation marker Ki-67 in the adult lung (Fig. 1B). Ki-67 was detected in nuclei of alveolar septal wall cells consistent with the location of AT1 or capillary endothelial cells. It was rarely detected in distal airway epithelial cells or large blood or lymphatic vessels. Due to the difficulty of distinguishing squamous-shaped AT1 from capillary endothelial cells using immunohistochemistry, flow cytometry was used to identify and quantify cells expressing Ki-67. Cells were isolated from the lungs of adult mice exposed to room air or hyperoxia and stained for Ki-67 and CD45/Ly-5 (leukocytes), CD326/EpCAM (epithelial cells), CD31/PECAM-1 or CD144/VE-cadherin (endothelial cells), and HOPX (AT1 cells). Approximately 10% of cells isolated from the lungs of mice exposed to room air expressed Ki-67. Neonatal hyperoxia increased the number of cells expressing Ki-67 by approximately threefold (Fig. 1C). Further analysis revealed Ki-67 was detected in ~20% of AT1 cells defined as CD326$^+$; HOPX$^+$ double-positive cells (Fig. 1D). Neonatal hyperoxia increased the number of adult CD326$^+$; HOPX$^+$ cells expressing Ki-67 by approximately two- to threefold such that Ki-67 was detected in about 50% of AT1 cells. Ki-67 was also detected in a small number of CD31$^+$ and CD144$^+$ endothelial cells, but neonatal hyperoxia did not affect the number of adult endothelial cells expressing Ki-67 (Fig. EV1A,B). Ki-67 was rarely detected in CD45$^+$ leukocytes of adult mice exposed previously to room air or hyperoxia as neonates. Since the FACS strategies used could not detect AT2 cells, Ki-67 immunostaining was conducted on adult Sftpc$^{EGFP}$ mice that express enhanced green fluorescent protein (EGFP) in all AT2 cells (Lo et al, 2008). Ki-67 was detected in ~1–2% of EGFP$^+$ AT2 cells in adult mice exposed to room air, with similar numbers seen in mice exposed to hyperoxia as neonates (Fig. EV1C).

To confirm that hyperoxia specifically increases Ki-67 in AT1 cells, AT1 cells were labeled with EGFP using Aqp5$^{Cre}$; Rosa26$^{mTmG}$ mice (Flodby et al, 2010). As expected, EGFP was widely detected in T1$^+$ alveolar squamous cells of adult mice exposed to room air or hyperoxia (Fig. 1E). Neonatal hyperoxia increased Ki-67 staining in alveolar cells, confirming that the effects of hyperoxia on Ki-67 expression were not specific for the C57BL/6J strain of mice. Lungs from adult Aqp5$^{Cre}$; Rosa26$^{mTmG}$ mice exposed to room air or hyperoxia as neonates were disassociated into single suspension, stained for Ki-67, and analyzed by flow cytometry. Ki-67 was detected in ~15% of EGFP-labeled AT1 cells from mice exposed to room air with an approximately three- to fourfold increase in mice that had been exposed to hyperoxia as neonates (Fig. 1E). Taken together, these findings reveal Ki-67 is expressed by some adult AT1 and AT2 cells, and that neonatal hyperoxia increases the number of adult AT1 cells expressing Ki-67.

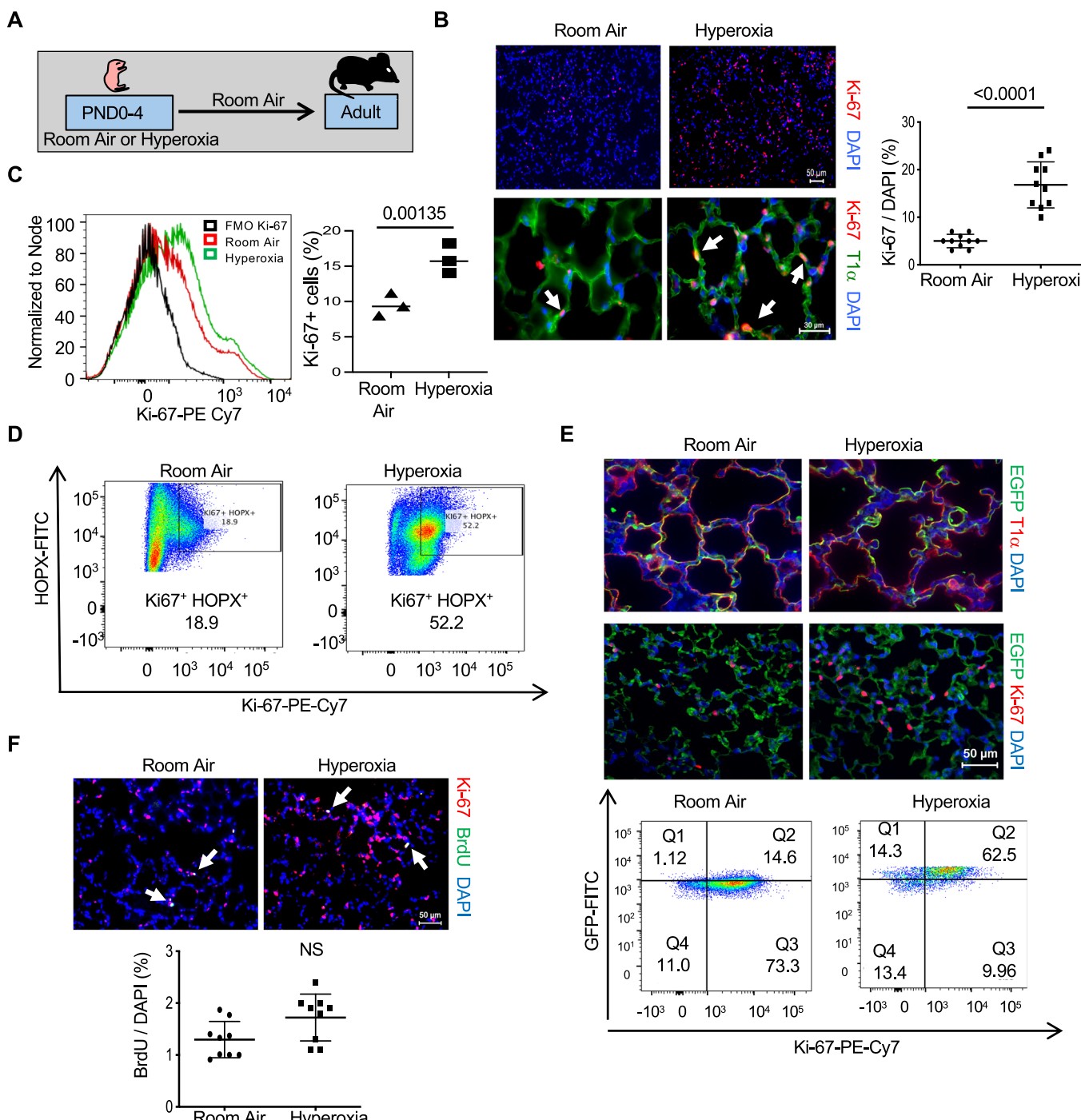

**Figure 1. Neonatal hyperoxia increases expression of Ki-67 in adult AT1 cells.**

(A) Cartoon of the experimental model. (B) Lungs of adult mice exposed to room air or hyperoxia as neonates were immunostained for Ki-67 (red), T1α (green), and counterstained with DAPI (blue). Arrows point to Ki-67+ cells. The proportion of Ki-67+ cells to DAPI+ nuclei were quantified and graphed. $n = 10$ mice per group. (Room air vs Hyperoxia: $P < 0.0001$). (C) Flow cytometric plots of Ki-67+ cells detected in disassociated lungs of adult mice exposed to room air or hyperoxia compared to FMO control. The proportion of Ki-67+ cells in 10,000 cells was quantified and graphed. $n = 3$ mice per group. (Room air vs Hyperoxia: $P = 0.00135$). (D) Flow cytometric plots of Ki-67+ and HOPX+ cells detected in lungs of adult mice exposed to room air or hyperoxia as neonates. The percentage of double-positive cells in 10,000 cells is shown in the embedded box. (E) Lungs of adult $Aqp5^{Cre}$; $Rosa26^{mTmG}$ mice exposed to room air or hyperoxia as neonates were immunostained for EGFP (green) and T1α (red) or EGFP (green) and Ki-67 (red) with DAPI (blue) counterstaining. FACS plots of Ki-67+ and EGFP+ cells from $Aqp5^{Cre}$; $Rosa26^{mTmG}$ exposed to room air or hyperoxia as neonates. (F) Lungs of adult mice exposed to room air or hyperoxia as neonates were immunostained for Ki-67 (red), BrdU (green), and counterstained with DAPI. Arrows point to BrdU+ cells. The proportion of BrdU+ cells to DAPI+ nuclei was graphed. $n = 9$ mice per group. NS not significant. Data in (B, C, F) are graphed as mean ± standard deviation with individual values shown as circles or squares. Scale bar in (B) = 30 μm and (E, F) = 50 μm. Data reflect biological replicates analyzed by Student's $t$ test in (B, C, F). Source data are available online for this figure.

The high number of AT1 cells expressing Ki-67 was surprising because AT1 cells do not proliferate. To test whether the increase in Ki-67-positive cells reflects an increase in cell proliferation, adult mice exposed to room air or hyperoxia as neonates were injected with BrdU, a thymidine analog that is incorporated into DNA during S phase. In contrast to Ki-67, ~1–2% of alveolar cells were labeled with BrdU, and there was no significant difference between mice previously exposed to room air or hyperoxia (Fig. 1F). We also analyzed the phosphorylation of histone H3 (Ser10), a mitotic marker. Phospho-H3 (Ser10) labeled cells were also rarely detected in lungs of adult mice exposed to room air with no significant increase in lungs of mice previously exposed to hyperoxia (Fig. EV1D). Thus, the majority of Ki-67-expressing cells in mice previously exposed to hyperoxia are not actively proliferating.

## AT1 cells expressing Ki-67 are derived from AT2 cells

Ki-67 is expressed by proliferating cells and cells that recently exited the cell cycle (Miller et al, 2018; Sobecki et al, 2017; van Oijen et al, 1998). Neonatal hyperoxia rapidly stimulates the proliferation of AT2 cells that are slowly depleted when mice return to room air (Yee et al, 2006). Since their lungs also express higher levels of the AT1 marker T1α, we speculated that Ki-67 present in adult AT1 cells may have been derived from neonatal AT2 cells that proliferated during hyperoxia and then differentiated into an AT1 cell after hyperoxia. To test this idea, $Sftpc^{CreERT}$; $Rosa26^{mTmG}$ mice were administered tamoxifen at birth to label AT2 cells with EGFP (Fig. 2A). The mice were then exposed to room air or hyperoxia until PND4, and then all mice were exposed to room air until they were 8 weeks (postnatal day 56) old. Lungs were then fixed and stained for EGFP and SFTPC used to identify AT2 cells or T1α used to identify AT1 cells. As previously shown (Yee et al, 2016), EGFP-labeled SFTPC+ AT2 cells were detected in lungs of PND4 mice exposed to room air, with significantly more cells present in lungs exposed to hyperoxia (Fig. 2B). EGFP-labeled SFTPC+ AT2 cells were also seen in adult lungs (PND56), however, EGFP staining was also detected in SFTPC-negative squamous alveolar cells of mice that had been exposed to hyperoxia. Squamous EGFP$^+$ cells expressing T1α in adult mice reflect AT1 cells derived from AT2 cells labeled with EGFP at birth. Flow cytometry was then used to confirm that these EGFP$^+$ squamous cells were AT1 cells and determine whether they express Ki-67. AT1 cells were identified by their expression of the homeodomain-only protein X (HOPX) transcription factor because T1α staining did not work well by flow cytometry. EGFP was detected in ~40% of HOPX+ AT1 cells of mice exposed to hyperoxia but rarely detected in HOPX+ cells of mice exposed to room air (Fig. 2C). We then investigated whether EGFP+; HOPX+ cells also express Ki-67 and found this this was true in cells of mice exposed to hyperoxia but not mice exposed to room air (Fig. 2D). Thus, neonatal hyperoxia increased the number of adult AT1 cells expressing Ki-67 derived from AT2 cells that may have undergone proliferation during hyperoxia.

## Ki-67 is not required for lung homeostasis

$Mki67^{CreERT}$ mice were mated to $R26^{mTmG}$ mice so that we could label Ki-67$^+$ cells with EGFP and track their fate over time. Tamoxifen was administered to adult $Mki67^{CreERT}$; $Rosa26^{mTmG}$ mice that were exposed to room air or hyperoxia as neonates (Fig. 3A). Lung, trachea, and gut were

harvested 7 days later and stained for EGFP. EGFP$^+$ cells were detected in proliferating crypt cells of the small intestine and in basal cells of the trachea but were surprisingly not detected in the lung (Fig. 3B). This suggests that the Ki-67 staining in the lung is not due to active Ki-67 transcription, consistent with the almost complete absence of proliferation in the adult lung. Importantly, the Ki-67 Cre driver hypomorphically inactivates Ki-67 expression (personal communication with Onur Basak and Hans Clevers, who created the mice). To confirm that $Mki67^{CreERT}$ mice have downregulated Ki-67 expression in the lung, its expression was evaluated by immunostaining the lungs of wildtype ($Mki67^{WT}$), Ki-67 heterozygous for CreERT ($Mki67^{WT/CreERT}$) and Ki-67 homozygous for CreERT ($Mki67^{CreERT}$) mice that had been exposed to room air or hyperoxia between PND0-4. As expected, Ki-67$^+$ cells were detected in alveoli of adult $Mki67^{WT}$ mice exposed to room air, with significantly more Ki-67$^+$ cells in mice exposed to hyperoxia (Fig. 3C). In contrast, fewer Ki-67$^+$ cells were seen in lungs of $Mki67^{WT/CreERT}$ mice that harbor one $CreERT$ allele in the Ki-67 3-untranslated region (3'-UTR), where few Ki-67$^+$ cells were seen in lungs of homozygous $Mki67^{CreERT}$ mice. Flow cytometry confirmed that the number of Ki-67$^+$ cells detected in $Mki67^{CreERT}$ mice was barely above background signal obtained with FMO controls (Fig. 3D). Although neonatal hyperoxia increased Ki-67 mRNA abundance in the adult lung, it had no effect on Ki-67 mRNA in Mki67$^{CrERT}$ mice, supporting the conclusion that the CreERT insertion disrupted expression of Ki-67 (Fig. 3E). In contrast to the lung, Ki-67 staining was still detected in the small intestine, possibly because cell proliferation is higher in this tissue (Fig. 3C, inset). This may also reflect translation from a downstream start site, compensating for the hypomorphism of the allele, such as previously seen in the intestinal epithelium of homozygous $Mki67$ TALEN-mutant mice (Sobecki et al, 2016). Despite loss of Ki-67, the morphology of lungs of $Mki67^{CreERT}$ mice exposed to room air appeared normal, and neonatal hyperoxia still caused alveolar simplification and depletion of AT2 cells (Fig. EV2). These findings show that normal Ki-67 expression is not required for lung development or homeostasis, which is consistent with another hypomorphic Ki-67 mouse line (Sobecki et al, 2016), nor for the ability of short-term hyperoxia to disrupt postnatal lung development.

## Ki-67 is required for the enhanced severity of respiratory viral infections in mice exposed to hyperoxia

Emerging evidence shows that Ki-67 is not required for cell proliferation but rather shapes the intensity of gene expression through interactions with chromatin regulators (Andres-Sanchez et al, 2022). Since neonatal hyperoxia enhances the severity of IAV infections, we hypothesized that high levels of Ki-67 might modulate the intensity of inflammatory gene expression during infection. Adult $Mki67^{WT}$ and $Mki67^{CreERT}$ hypomorph mice exposed to room air or hyperoxia as neonates were infected with a sublethal dose of IAV strain Hkx31 (H3N2). This strain of IAV was chosen because it causes mild lung injury in mice exposed to room air but severe ALI and mortality in mice exposed to hyperoxia (Brandes et al, 2013; O'Reilly et al, 2008). As expected, infected $Mki67^{WT}$ mice exposed to hyperoxia as neonates exhibited greater mortality and weight loss than infected wild-type mice exposed to room air (Fig. 4A,B). By post-infection day 14, lungs of infected $Mki67^{WT}$ mice exposed to hyperoxia contained evidence of persistent inflammatory cells and fibrotic honeycombing (Fig. 4C). Keratin 5 (Krt5) positive basal cells seen under conditions of severe lung injury were detected underlying the distal airway epithelium of infected $Mki67^{WT}$ exposed to room air, with additional staining

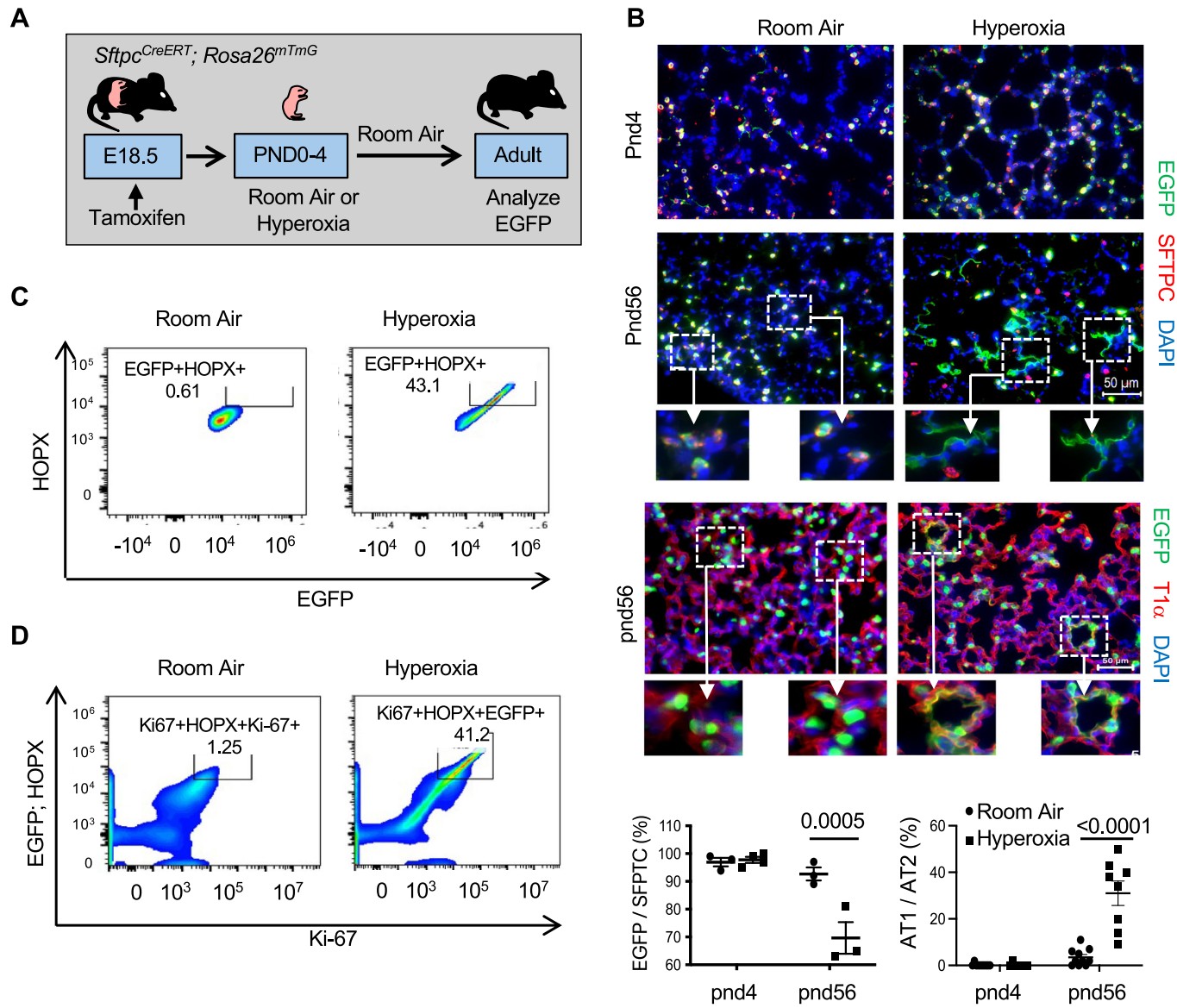

**Figure 2. Neonatal hyperoxia stimulates differentiation of AT2 to AT1 cells that express Ki-67.**

(A) Cartoon showing the experimental model of exposing mice to hyperoxia and returning to room air. (B) Lungs of PND4 and adult $Sftpc^{CreERT}$; $Rosa26^{mTmG}$ mice exposed to room air or hyperoxia as neonates were immunostained for EGFP (green), SFTPC (red) or T1α (red), and counterstained with DAPI (blue). Hatched inset boxes are enlarged below individual images. The proportion of EGFP+ and SFTPC+ cells was quantified and graphed. $n = 3$ (RA PND4), 5 (O2 PND4), 3 (RA PND56), and 3 (O2 PND56). (Room Air vs Hyperoxia at PND56: $P = 0.0005$). The proportion of squamous EGFP+ to cuboidal EGFP+ staining was quantified and graphed. $n = 10$ (RA PND4), 10 (O2 PND4), 8 (RA PND56), and 8 (O2 PND56). (Room air vs Hyperoxia at PND56: $P < 0.0001$). (C) FACs analysis of EGFP+ and HOPX+ cells isolated from lungs of adult $Sftpc^{CreERT}$; $Rosa26^{mTmG}$ mice exposed to room air or hyperoxia as neonates. (D) FACS analysis of Ki-67+ cells that also express EGFP and HOPX. Data in (B) are graphed as mean ± standard deviation with individual samples shown as circles and squares. Scale bar in (B) = 50 μm and in (C) = 200 μm. Data reflect biological replicates analyzed by one-way ANOVA using Tukey-Kramer HSD in (B). Source data are available online for this figure.

detected in pod-like clusters of $Mki67^{WT}$ exposed to neonatal hyperoxia (Fig. 4D). These alveolar pod structures also stained positive for keratin 8 (Krt8), an airway-specific intermediate filament expressed by transitional AT2-AT1 cells that also accumulate under conditions of severe lung injury (Fig. 4E) (Strunz et al, 2020).

In contrast, $Mki67^{CreERT}$ hypomorph mice responded differently to IAV. Although these mice also lost a little weight during infection, they displayed no mortality regardless of whether they had been exposed to room air or hyperoxia as neonates (Fig. 4A,B).

On post-infection day 14, lungs of infected $Mki67^{CreERT}$ hypomorph mice exposed to room air or hyperoxia appeared histologically normal (Fig. 4C). Krt5+ basal cells were observed underlying the airway epithelium of infected $Mki67^{CreERT}$ hypomorph mice but rarely in the alveolar space (Fig. 4D). Krt8 expression was detected in the airway epithelium but not in the alveolar region suggesting lack of AT2-AT1 transitional cells (Fig. 4E). Thus, loss of Ki-67 protects mice exposed to hyperoxia as neonates from the severe effects of Hkx31 viral infection.

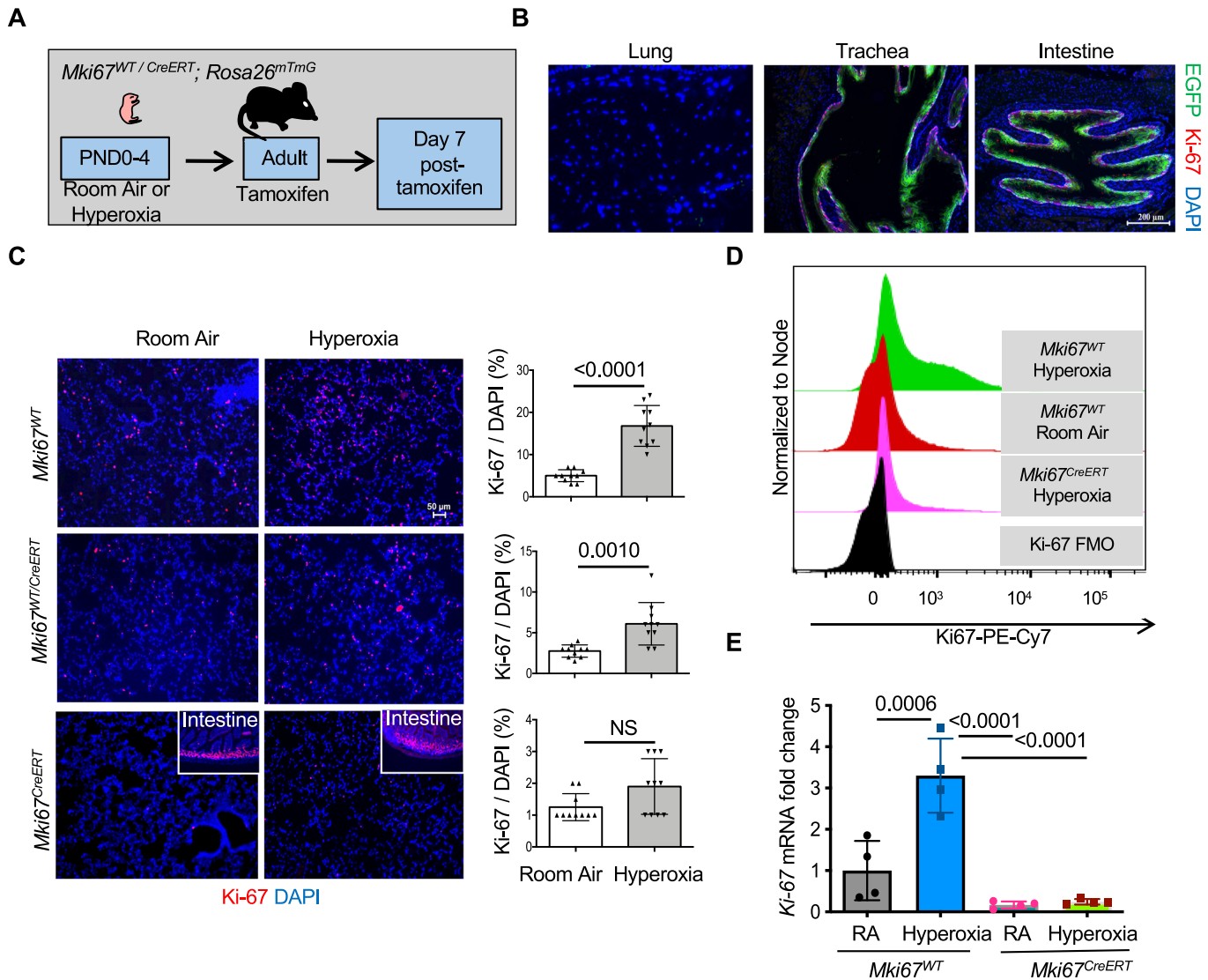

**Figure 3. Mki67^CreERT mice are Ki-67 hypomorphs in the lung.**

(A) Cartoon showing the experimental plan of exposing *Mki67^CreERT*; *Rosa26^mTmG* to room air or hyperoxia as neonates, followed by tamoxifen as adults. (B) Lungs, trachea, and intestine from adult *Mki67^CreERT*; *Rosa26^mTmG* administered tamoxifen were stained for EGFP (green), Ki-67 (red), and DAPI (blue). (C) Lungs of adult *Mki67^WT*, *Mki67^WT/CreERT*, or *Mki67^CreERT* mice exposed to room air or hyperoxia as neonates were stained for Ki-67 (red) and DAPI (blue). The proportion of Ki-67^+ cells were quantified and graphed. n = 10 mice per group. (*Mki67^WT*: Room air vs Hyperoxia: $P < 0.0001$; *Mki67^WT/CreERT*: Room air vs Hyperoxia: $P = 0.0010$; *Mki67^CreERT*: Room air vs Hyperoxia: NS=not significant). (D) FACS plot of Ki-67^+ cells detected in lungs of adult *Mki67^WT* exposed to room air or hyperoxia for comparison against *Mki67^CreERT* exposed to hyperoxia and FMO control after normalizing to NODE. (E) Expression of Ki-67 mRNA was determined in lungs of adult *Mki67^WT* and *Mki67^CreERT* mice exposed to room air or hyperoxia as neonates, normalized to 18S RNA, and graphed relative to *Mki67^WT* in room air. n = 4 mice per group. (*Mki67^WT* Room air vs *Mki67^WT* Hyperoxia: $P = 0.0034$; *Mki67^WT* Hyperoxia vs *Mki67^CreERT* Room air: $P < 0.0001$; *Mki67^WT* Hyperoxia vs *Mki67^CreERT* Hyperoxia: $P < 0.0001$). Data in (C, E) are graphed as mean ± standard deviation with individual samples shown as triangles, circles, or squares. Scale bar in (B) = 50 μm. Data reflect biological replicates analyzed by one-way ANOVA using Tukey-Kramer HSD in (C, E). Source data are available online for this figure.

On post-infection day 5, large patches of TUNEL-positive apoptotic debris were detected in *Mki67^WT* mice exposed to hyperoxia (Fig. EV3). These apoptotic cells were likely AT1 cells because they also stained positive for T1α and were often associated with infiltrating neutrophils identified by NETosis due to increased citrullinated histone H3 staining (Fig. 4F). This pathology was reduced in infected *Mki67^WT* mice exposed to room air and rarely seen in *Mki67^CreERT* hypomorph mice exposed to room air or hyperoxia.

Seeing evidence of neutrophil infiltration motivated us to quantify the number and type of leukocytes in the bronchoalveolar lavage fluid of mice before and 1, 3, 5, and 7 days after infection. Prior to infection, the total number of leukocytes was similar in the lungs of *Mki67^WT* and *Mki67^CreERT* hypomorph mice exposed to room air or hyperoxia (Fig. 5A,B). Most of these cells were macrophages. The total number of leukocytes increased after infection and was significantly faster and greater in *Mki67^WT* mice exposed to hyperoxia (Fig. 5A). This large increase in leukocytes

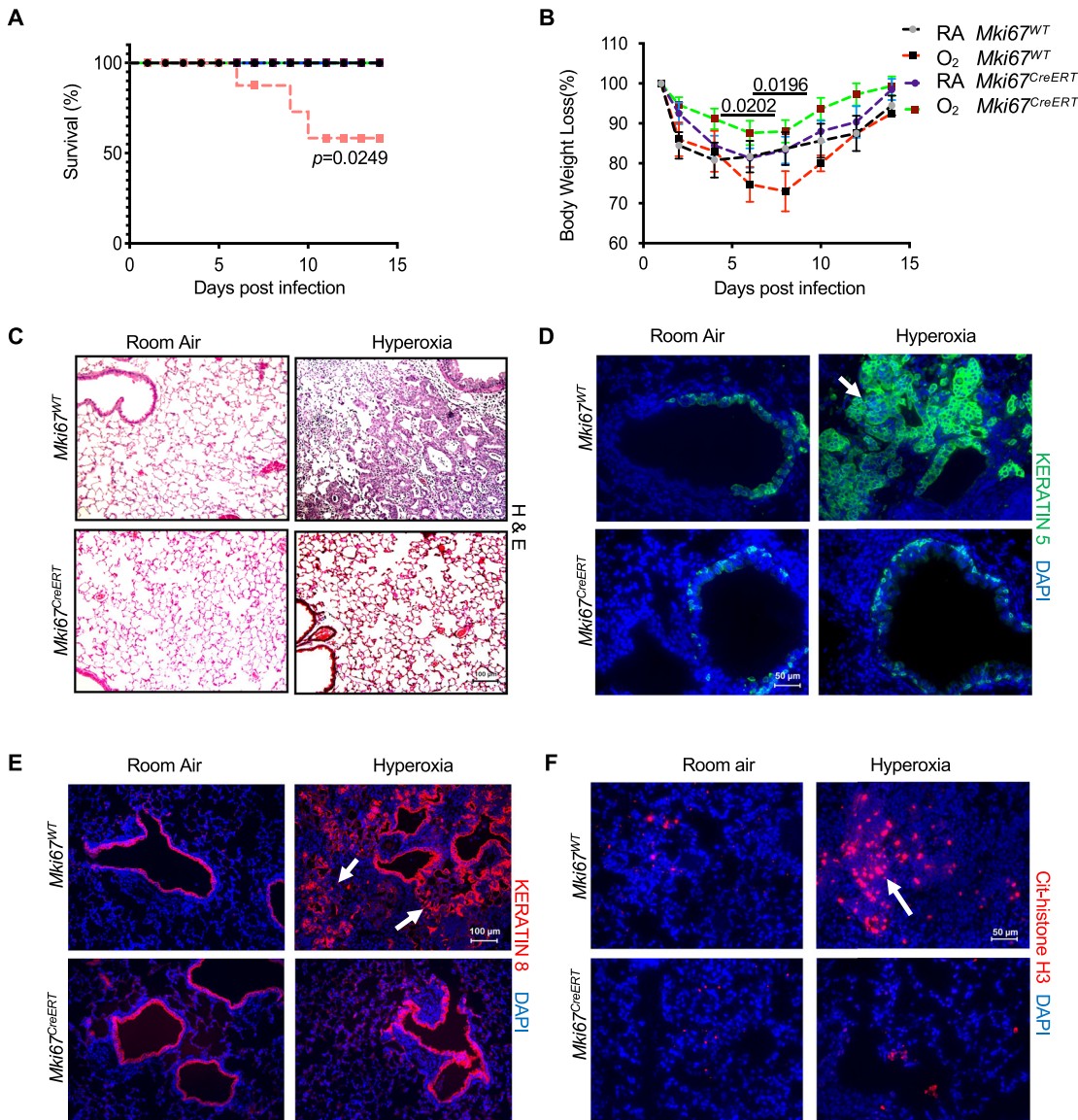

**Figure 4. Neonatal hyperoxia increases the severity of IAV infections in *Mki67^WT* but not *Mki67^CreERT* hypomorph mice.**

Adult *Mki67^WT* and *Mki67^CreERT* exposed to room air or hyperoxia as neonates were infected with Hkx31 strain of IAV. (A) Neonatal hyperoxia significantly reduced the survival of infected *Mki67^WT* mice but not *Mki67^CreERT* mice. n = 10 per group. (B) Neonatal hyperoxia significantly increased weight loss of infected *Mki67^WT* mice but not *Mki67^CreERT* hypomorph mice. n = 10 mice per group. (Day 6: O₂ Mki67^CreERT vs RA Mki67^WT: P = 0.0202; Day 8: O₂ Mki67^CreERT vs RA Mki67^WT: P = 0.0.0196). (C–F) Lungs harvested on post-infection day 14 were stained with (C) Hematoxylin and Eosin to visualize pathology, (D) keratin 5 used to detect basal cells, (E) keratin 8 used to detect alveolar epithelial transitional cells, and (F) citrullinated histone H3 used to identify NETotic neutrophils. Arrows in (D–F) point respectively to keratin 5⁺ pods, keratin 8⁺ transitional cells, and citrullinated H3⁺ neutrophils. Data in (B) is graphed as mean weight loss after infection ± standard error of the mean. Scale bar in (C, E) = 50 μm and in (D, F) = 100 μm. Data reflects biological replicates analyzed by Log-Rank (Mantel–Cox) test in (A) and Student's *t* test comparing individual post-infection days (B). Source data are available online for this figure.

was primarily due to a rapid influx of neutrophils within the first 24 h of infection, which then progressively cleared such that few neutrophils were detected by post-infection day 7 (Fig. 5C). This large increase in neutrophils reduced the proportion of macrophages detected through post-infection day 5 (Fig. 5B) and impacted the proportion of lymphocytes detected on post-infection days 3 and 7 (Fig. 5D). In contrast, neutrophils were not excessively recruited when *Mki67^CreERT* hypomorph mice exposed to room air or hyperoxia were infected with virus

(Fig. 5C,E). The proportion of macrophages and neutrophils was also less impacted in these mice (Fig. 5B,D). Thus, neonatal hyperoxia increases, in a Ki-67-dependent manner, the recruitment of neutrophils during infection that may drive excessive death of AT1 cells as they undergo NETosis.

To explore the mechanism for the Ki-67 dependency of neutrophils, we evaluated expression of mRNAs encoding pro-inflammatory cytokines and chemokines in lungs of *Mki67^WT* and *Mki67^CreERT* hypomorph mice exposed to room air or hyperoxia.

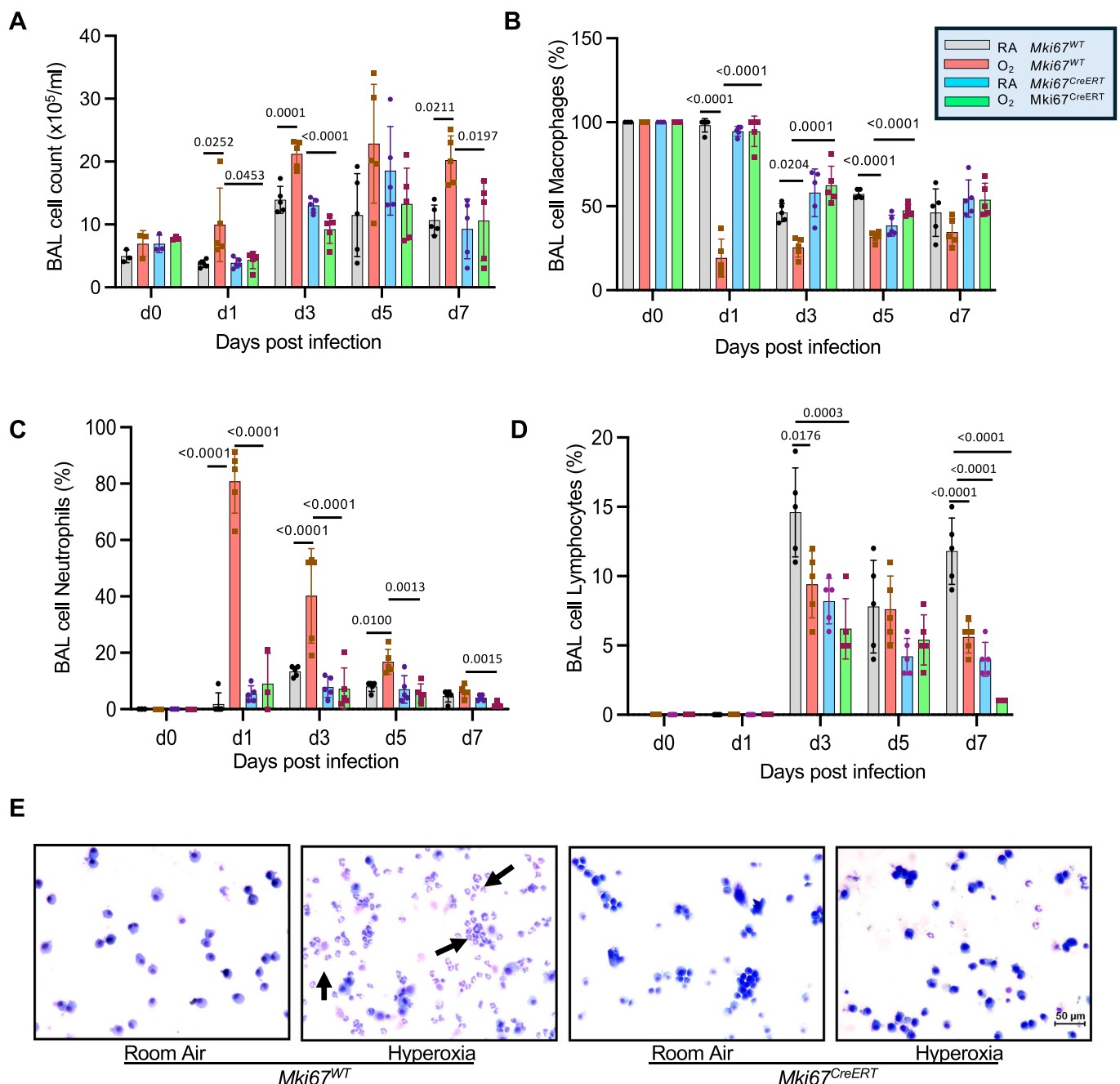

**Figure 5. Neonatal hyperoxia enhances the recruitment of neutrophils in the lungs of infected *Mki67^WT* but not *Mki67^CreERT* hypomorph mice.**

Bronchoalveolar lavage washes were performed on adult *Mki67^WT* and *Mki67^CreERT* exposed to room air or hyperoxia before and after infection with Hkx31 IAV. (A) The total number of leukocytes, (B) the proportion of macrophages, (C) the proportion of neutrophils, and (D) the proportion of lymphocytes were quantified and graphed *n* = 5 mice per group. (A: D1: RA *Mki67^WT* vs O₂ *Mki67^WT*: P = 0.0252; O₂ *Mki67^WT* vs *Mki67^CreERT*: P = 0.0453; D3: RA *Mki67^WT* vs O₂ *Mki67^WT*: P = 0.0001; O₂ *Mki67^WT* vs *Mki67^CreERT*: P < 0.0001; d7: RA *Mki67^WT* vs O₂ *Mki67^WT*: P = 0.0211; O₂ *Mki67^WT* vs *Mki67^CreERT*: P = 0.0197). (B: D1: RA *Mki67^WT* vs O₂ *Mki67^WT*: P < 0.0001; O₂ *Mki67^WT* vs *Mki67^CreERT*: P < 0.0001; D3: RA *Mki67^WT* vs O₂ *Mki67^WT*: P = 0.024; O₂ *Mki67^WT* vs *Mki67^CreERT*: P = 0.0001; d5: RA *Mki67^WT* vs O₂ *Mki67^WT*: P < 0.0001; O₂ *Mki67^WT* vs *Mki67^CreERT*: P < 0.0001). (C: D1: RA *Mki67^WT* vs O₂ *Mki67^WT*: P < 0.0001; O₂ *Mki67^WT* vs *Mki67^CreERT*: P < 0.0001; D3: RA *Mki67^WT* vs O₂ *Mki67^WT*: P < 0.0001, O₂ *Mki67^WT* vs *Mki67^CreERT*: P < 0.0001; d5: RA *Mki67^WT* vs O₂ *Mki67^WT*: P = 0.0100; O₂ *Mki67^WT* vs *Mki67^CreERT*: P = 0.0013; D7: O₂ *Mki67^WT* vs *Mki67^CreERT*: P = 0.0015). (E) Representative images of cytospins collected from mice on post-infection day 1. Arrows point to neutrophils in lavage collected from infected *Mki67^WT* mice exposed to hyperoxia Data in (A–D) is graphed as mean ± standard deviation with individual mice shown in circles or squares. Scale bar in (E) = 50 μm. Data reflect biological replicates analyzed by one-way ANOVA using Tukey-Kramer HSD in (A–D). Source data are available online for this figure.

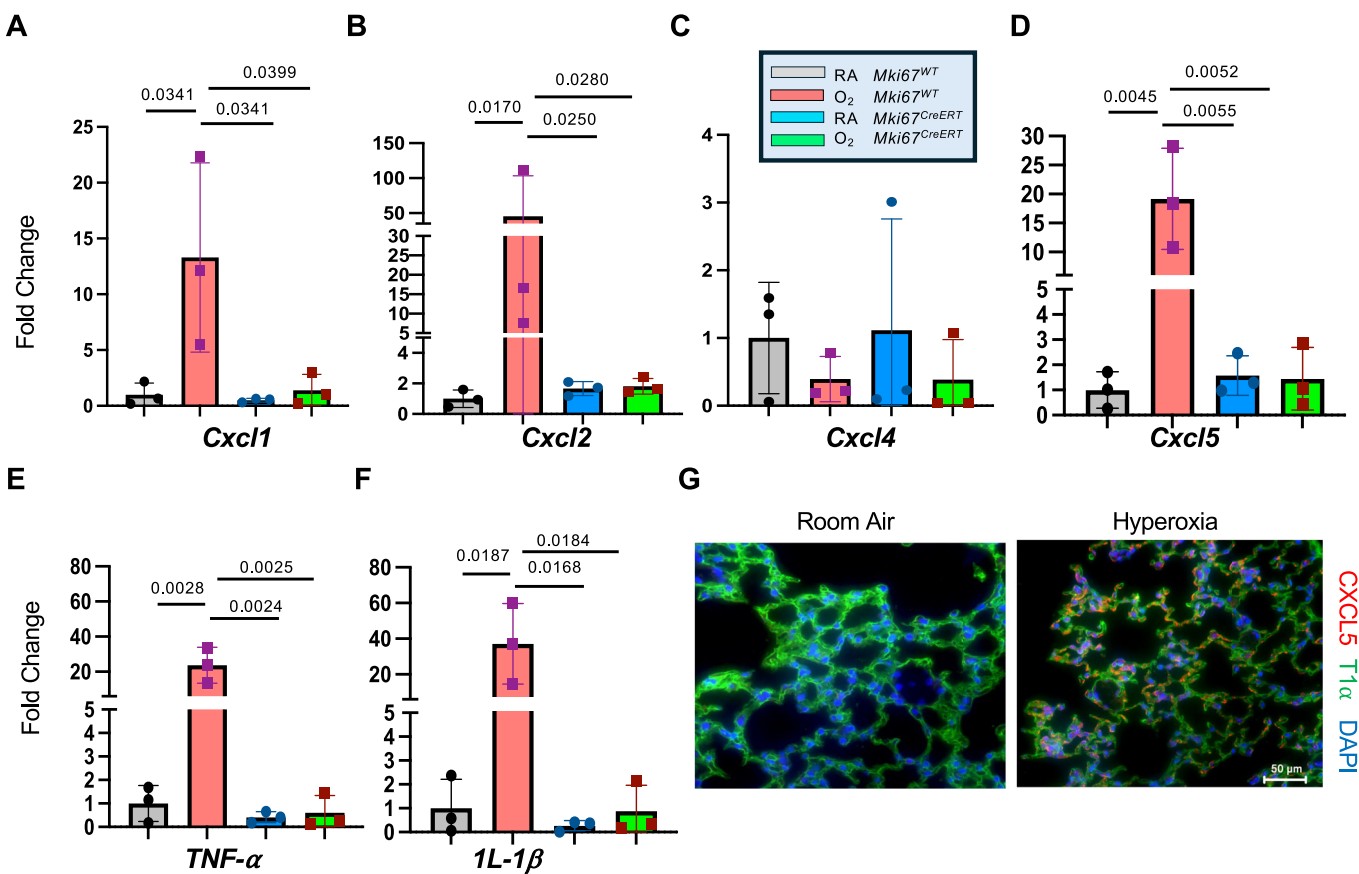

**Figure 6. Neonatal hyperoxia increase expression of CXCL chemokines in infected _Mki67^WT_ but not _Mki67^CreERT_ hypomorph mice.**

Adult _Mki67^WT_ and _Mki67^CreERT_ exposed to room air or hyperoxia as neonates were infected with Hkx31 IAV. (A–F) The expression of _Cxcl1, Cxlc2, Cxcl4, Cxcl5,_ TNF-α, and _IL-1β_ in lungs one day after infection was measured by qRT-PCR and graphed. $n = 3$ mice per group (_Cxcl1_: RA _Mki67^WT_ vs O$_2$ _Mki67^WT_: $P = 0.0341$; O$_2$ _Mki67^WT_ vs RA _Mki67^CreERT_: $P = 0.0341$; O$_2$ _Mki67^WT_ vs O$_2$ _Mki67^CreERT_: $P = 0.0399$). (_Cxcl2_: RA _Mki67^WT_ vs O$_2$ _Mki67^WT_: $P = 0.0170$; O$_2$ _Mki67^WT_ vs RA _Mki67^CreERT_: $P = 0.0250$; O$_2$ _Mki67^WT_ vs O$_2$ _Mki67^CreERT_: $P = 0.0280$). (_Cxcl5_: RA _Mki67^WT_ vs O$_2$ _Mki67^WT_: $P = 0.0045$; O$_2$ _Mki67^WT_ vs RA _Mki67^CreERT_: $P = 0.0055$; O$_2$ _Mki67^WT_ vs O$_2$ _Mki67^CreERT_: $P = 0.0052$). (TNF-α: RA _Mki67^WT_ vs O$_2$ _Mki67^WT_: $P = 0.0028$; O$_2$ _Mki67^WT_ vs RA _Mki67^CreERT_: $P = 0.0024$; O$_2$ _Mki67^WT_ vs O$_2$ _Mki67^CreERT_: $P = 0.0025$). (_IL-1β_: RA _Mki67^WT_ vs O$_2$ _Mki67^WT_: $P = 0.0187$; O$_2$ _Mki67^WT_ vs RA _Mki67^CreERT_: $P = 0.0168$; O$_2$ _Mki67^WT_ vs O$_2$ _Mki67^CreERT_: $P = 0.0184$). (G) Lungs of infected _Mki67^WT_ mice exposed to room air or hyperoxia were immunostained for CXCL5 (red), T1α (green) and counterstained with DAPI (blue). Data in (A–F) is graphed as mean ± standard deviation with individual mice shown as circles or squares. Scale bar in (G) = 50 μm. Data reflect biological replicates analyzed by one-way ANOVA using Tukey-Kramer HSD (A–F). Source data are available online for this figure.

One day after infection, the expression of _Cxcl1, Cxcl2, Cxcl5,_ TNF-α, and _IL-1β_ were 15 – 50-fold higher in lungs of _Mki67^WT_ mice exposed to hyperoxia when compared to _Mki67^WT_ mice exposed to room air (Fig. 6A–F). Increased CXCL5 staining was also detected in AT1 cells defined by expression of T1α (Fig. 6G). These changes seemed to reflect inflammatory signals driving neutrophil recruitment because hyperoxia did not increase expression of _Cxcl4_, a chemokine that stimulates platelets (Fig. 6C). In contrast to _Mki67^WT_ mice, the expression of inflammatory cytokine and chemokines were low in infected _Mki67^CreERT_ mice exposed to room air or hyperoxia as neonates.

## Restoring Ki-67 in the lung enhances neutrophil recruitment and the pathogenicity of IAV

To investigate whether Ki-67 in the lung is directly involved in gene expression changes that enhance neutrophil recruitment, we restored Ki-67 expression in _Mki67^CreERT_ mice by electroporation-mediated gene delivery, and then assessed neutrophil recruitment when mice were infected with IAV. We expressed a human Ki-67-mCherry fusion protein under control of the ubiquitous cytomegalovirus (CMV) promoter from a vector (pcDNA3.1) containing SV40 enhancer sequences (Dean, 1997). The pcDNA3.1-Ki-67mCherry and parent pcDNA3.1 plasmids were instilled intratracheally into the lungs of naïve adult _Mki67^WT_ and _Mki67^CreERT_ hypomorph mice and trans-thoracically electroporated to facilitate cellular uptake of the plasmid. Mice were recovered for 48 h and then Ki-67 was evaluated by immunohistochemistry in one group of mice while the remainder were infected with Hkx31 IAV (Fig. 7A). Increased Ki-67 staining was consistently detected in airways and alveoli of mice transduced with pcDNA3.1-Ki-67mCherry plasmid (Fig. 7B). The number of Ki-67+ cells however varied between the mice, reflecting the variability of this approach for delivering genes to individual lobes and in different mice.

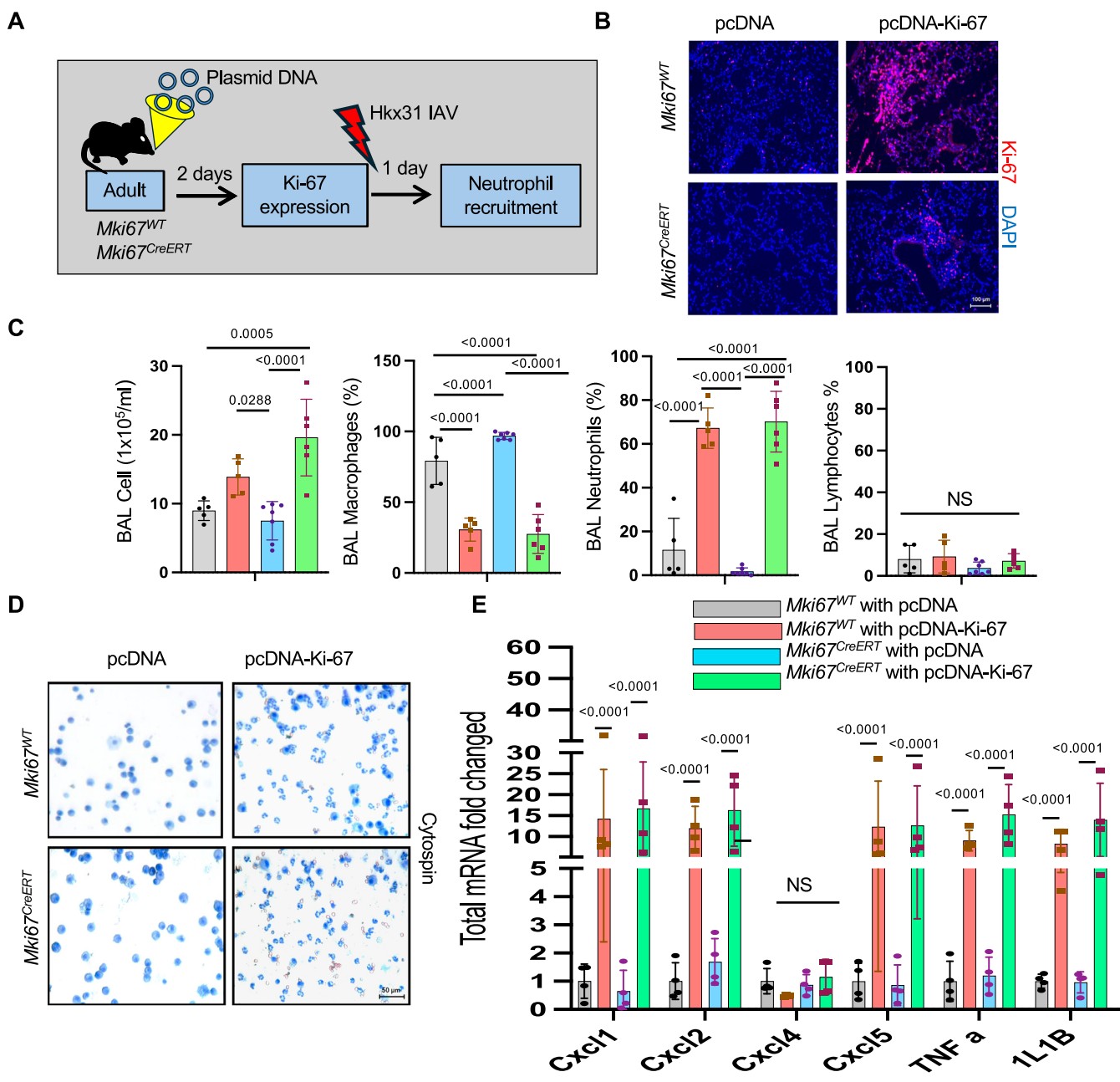

**Figure 7. Gene delivery of Ki-67 cDNA enhances neutrophil recruitment in *Mki67^WT* and *Mki67^CreERT* mice infected with Hkx31 IAV.**

(A) Cartoon showing the experimental plan for delivering pcDNA or pcDNA-Ki-67 expression plasmid using electroporation to lungs of adult *Mki67^WT* and *Mki67^CreERT* mice via electroporation. (B) Lungs were collected 48 h after gene delivery and immunostained for Ki-67 (red) and counterstained with DAPI (blue). (C) Bronchoalveolar lavages were performed 24 h after gene-delivered *Mki67^WT* and *Mki67^CreERT* mice were infected with Hkx31. The total number of leukocytes, the proportion of macrophages, the proportion of neutrophils, and the proportion of lymphocytes were quantified and graphed. *n* = 5 *Mki67^WT* with pcDNA or pcDNA-Ki-67, 7 *Mki67^CreERT* with pcDNA, and 6 *Mki67^CreERT* with pcDNA-Ki-67. (BAL Cells: *Mki67^WT* with pcDNA vs *Mki67^CreERT* with pcDNA-Ki-67: *P* = 0.0005; *Mki67^WT* with pcDNA-Ki-67 vs *Mki67^CreERT* with pcDNA: *P* = 0.0288; *Mki67^CreERT* with pcDNA vs *Mki67^CreERT* with pcDNA-Ki-67: *P* < 0.0001). (BAL macrophages: *Mki67^WT* with pcDNA vs *Mki67^WT* with pcDNA-Ki-67: *P* < 0.0001; *Mki67^WT* with pcDNA vs *Mki67^CreERT* with pcDNA: *P* < 0.0001; *Mki67^WT* with pcDNA vs *Mki67^CreERT* with pcDNA-Ki-67: *P* < 0.0001; *Mki67^CreERT* with pcDNA vs *Mki67^CreERT* with pcDNA-Ki-67: *P* < 0.0001). (BAL neutrophils: *Mki67^WT* with pcDNA vs *Mki67^WT* with pcDNA-Ki-67: *P* < 0.0001; *Mki67^WT* with pcDNA vs *Mki67^CreERT* with pcDNA: *P* < 0.0001; *Mki67^WT* with pcDNA vs *Mki67^CreERT* with pcDNA-Ki-67: *P* < 0.0001; *Mki67^CreERT* with pcDNA vs *Mki67^CreERT* with pcDNA-Ki-67: *P* < 0.0001). (BAL lymphocytes: NS = not significant). (D) Representative cytospin images. (E) qRT-PCR was used to assess expression of *Cxcl1*, *Cxlc2*, *Cxcl4*, *Cxcl5*, *TNF-α*, and *IL-1β* in lungs one day gene-delivered *Mki67^WT* and *Mki67^CreERT* mice were infected with Hkx31. mRNA expression was graphed as fold change relative to infected *Mki67^WT* mice transduced with control pcDNA plasmid. *n* = 4 mice per group. (*Cxcl1, Cxcl2, Cxcl5, TNF-α, IL-1β*: *Mki67^WT* with pcDNA vs *Mki67^WT* with pcDNA-Ki-67: *P* < 0.0001; *Mki67^CreERT* with pcDNA vs *Mki67^CreERT* with pcDNA-Ki-67: *P* < 0.0001; Cxcl5: NS = not significant). Data in (C, E) are graphed as mean ± standard deviation with individual mice shown as circles or squares. Scale bar in (B) = 100 μm and (D) = 50 μm. Data reflect biological replicates analyzed by one-way ANOVA using Tukey-Kramer HSD (C, E). Source data are available online for this figure.

The number and identity of leukocytes were then determined in the lavage fluid collected one day after infection with IAV. Ectopic expression of Ki-67 in Mki67^WT and *Mki67^CreERT* hypomorph mice significantly increased the number of leukocytes recruited to lungs infected with Hkx31 IAV (Fig. 7C,D). This reflected a large increase in the proportion of neutrophils. As expected, expressing Ki-67 did not impact the low number of lymphocytes recruited in these conditions. Given that infection was required for Ki-67 to stimulate neutrophil recruitment, we investigated whether Ki-67 affected the expression of chemokines stimulating neutrophil recruitment. Mice transduced with Ki-67mCherry plasmid and infected with IAV expressed significantly higher levels of *Cxcl1, Cxcl2, Cxcl5, TNF-α,* and *IL-1β* mRNA than infected mice transduced with the empty vector (Fig. 7E). These changes were not seen in uninfected mice. None of the mice expressed higher levels of *Cxcl4*, a chemokine responsible for recruiting platelets to the lung. Thus, expressing Ki-67 in the lung stimulates neutrophil recruitment following IAV infection.

## Depleting neutrophils reduces the severity of IAV infections seen in adult mice exposed to hyperoxia

To confirm that hyperoxia enhances the severity of IAV infections by stimulating the recruitment of neutrophils, adult *Mki67^WT* mice exposed to room air or hyperoxia as neonates were injected with 300 μg of anti-Lys6G (1A8) antibody to deplete neutrophils or rat IgG used as control (Fig. 8A). The dose of 1A8 antibody was chosen based upon a prior study showing that it protected mice infected with PR8 IAV (Brandes et al, 2013) and our pilot experiments with *Mki67^WT* mice showing that it suppresses neutrophil recruitment during infection (Fig. 8B). Adult *Mki67^WT* mice exposed to room air or hyperoxia as neonates were injected with anti-Lys6G and then infected with Hkx31 IAV. Depleting neutrophils by this approach improved survival and weights of *Mki67^WT* mice exposed to hyperoxia to levels seen in *Mki67^WT* mice exposed to room air (Fig. 8C,D). Depleting neutrophils also reduced lung scarring and fibrosis of infected mice exposed to hyperoxia (Fig. 8E). Neonatal

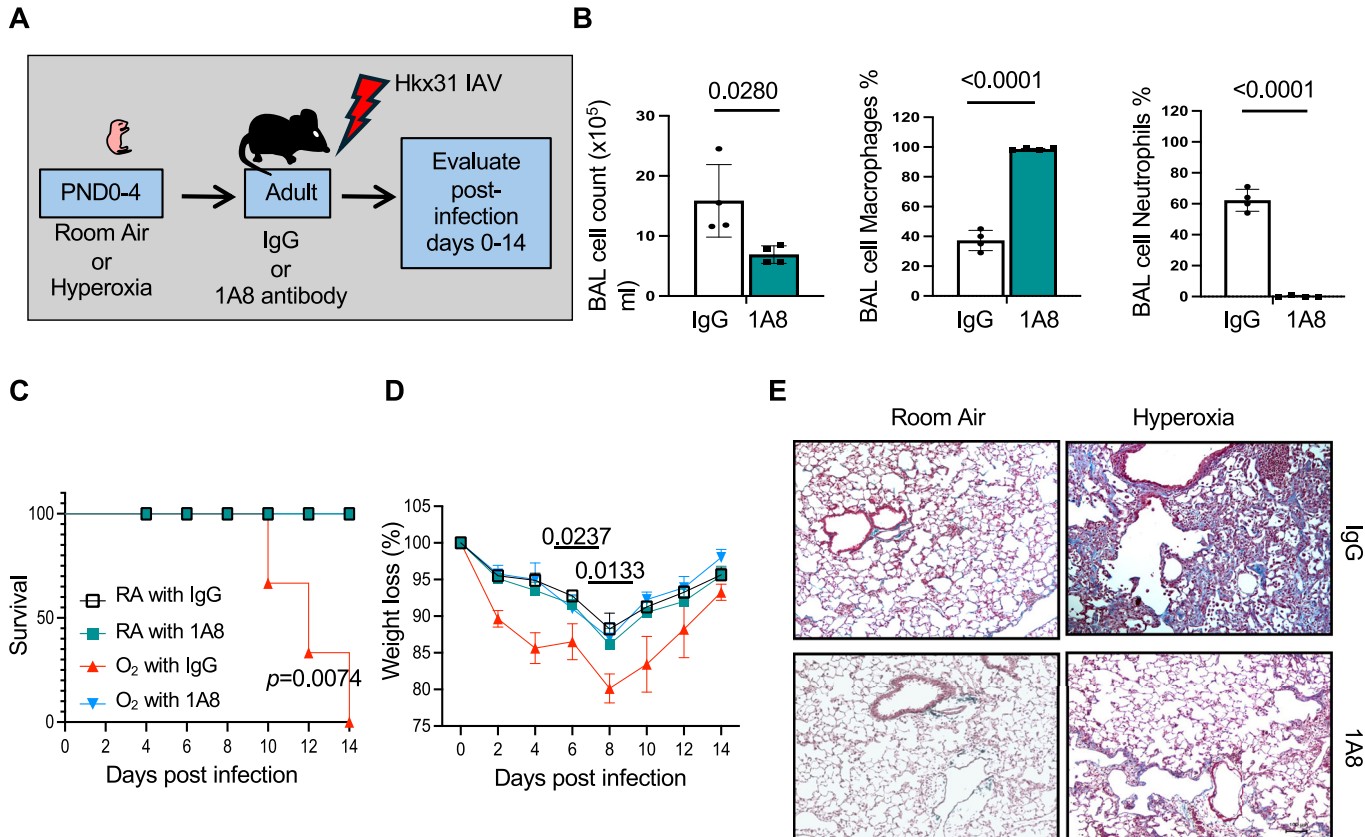

**Figure 8. Depleting neutrophils prevents acute lung injury in infected mice exposed to neonatal hyperoxia.**

(A) Cartoon showing the experimental plan for administering IgG or 1A8 antibody to adult mice exposed to room air or hyperoxia as neonates followed by infection with Hkx31 IAV. Bronchoalveolar lavage washes were performed on adult mice administered IgG or 1A8 and infected with IAV for one day. (B) The total number of leukocytes, the proportion of macrophages, and the proportion of neutrophils were quantified in room air mice on post-infection day 3 and graphed. $n = 4$ mice per group. (BAL cells: IgG vs 1A8: $P = 0.0280$; BAL macrophages: IgG vs 1A8: $P < 0.0001$; BAL neutrophils: IgG vs 1A8: $P < 0.0001$). (C) 1A8 antibody significantly improved the survival of infected mice exposed to hyperoxia as neonates. $n = 8$ mice per group. (O₂ with IgG vs RA with IgG, RA with IA8, or O₂ with IA8; $P = 0.0074$). (D) 1A8 antibody significantly reduced weight loss in infected mice exposed to neonatal hyperoxia. $n = 8$ mice per group. Average weight loss ± standard error of the mean relative to uninfected mice. (Day 6: O₂ IgG vs O₂ 1A8: $p = 0.0_237$; Day 8: O₂ IgG vs O₂ 1A8: $P = 0.0133$). (E) Trichrome staining of post-infected day 14 lungs obtained from mice administered IgG or 1A8. Data in (B) graphed as mean ± standard deviation with individual mice shown as circles or squares. Scale bar in (E) = 100 μm. Data reflects biological replicates analyzed by one-way ANOVA using Tukey-Kramer HSD (B), Log-Rank (Mantel–Cox) test in (C), and Student's *t* test comparing individual post-infection days (D). Source data are available online for this figure.

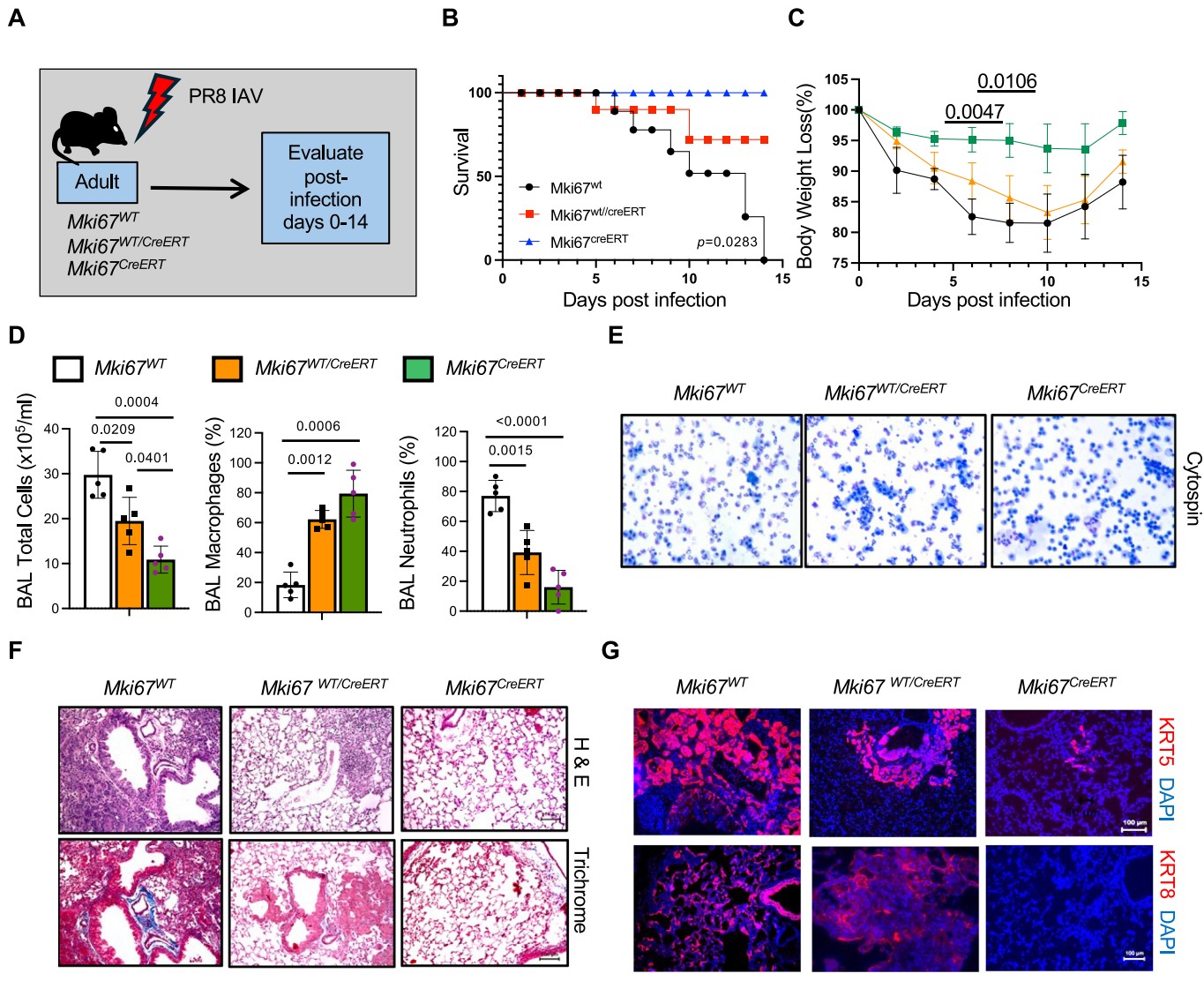

**Figure 9. Mki67^CreERT hypomorphs are tolerant to pathogenic PR8 IAV infections.**

(A) Cartoon model showing the experimental plan for infecting Mki67^WT, Mki67^WT/CreERT, and Mki67^CreERT with a lethal dose of PR8 IAV. (B) Survival of Mki67^WT, Mki67^WT/CreERT, and Mki67^CreERT mice infected with IAV. n = 8 mice per group. (Mki67^CreERT vs Mki67^WT/CreERT vs Mki67^WT: P = 0.0283). (C) Weight loss in Mki67^WT, Mki67^WT/CreERT, and Mki67^CreERT mice infected with IAV. n = 8 mice per group. (Day 6: Mki67^CreERT vs Mki67^WT: P = 0.0047; Day 8: Mki67^CreERT vs Mki67^WT: P = 0.0106). (D) Inflammatory cell number and the percentage of macrophages and neutrophils were determined in BAL washes from Mki67^WT, Mki67^WT/CreERT, and Mki67^CreERT mice on post-infection day 3. n = 5 mice per group. (BAL cells: Mki67^WT vs Mki67^WT/CreERT: P = 0.0209; Mki67^WT vs Mki67^CreERT: P = 0.0401; Mki67^WT/CreERT vs Mki67^CreERT: P = 0.0004). (BAL macrophages: Mki67^WT vs Mki67^WT/CreERT: P = 0.0012; Mki67^WT vs Mki67^CreERT: P = 0.0006). (BAL neutrophil: Mki67^WT vs Mki67^WT/CreERT: P = 0.0015; Mki67^WT vs Mki67^CreERT: P < 0.0001). (E) Representative cytospin images of BAL leukocytes collected from Mki67^WT, Mki67^WT/CreERT, and Mki67^CreERT mice on post-infection day 3. (F) Lung pathology of Mki67^WT, Mki67^WT/CreERT, and Mki67^CreERT mice on post-infection day on post-infection day 14. (G) Keratin 5 and keratin 8 staining in lungs of Mki67^WT, Mki67^WT/CreERT, and Mki67^CreERT mice on post-infection day 14. Data in (C) graphed a mean ± standard error of the mean and data in (D) graphed as mean ± standard deviation. Scale bar in (E–G) = 100 μm. Data reflect biological replicates analyzed by Log-Rank (Mantel–Cox) test in (B), Student's t test comparing individual post-infection days (C), and one-way ANOVA using Tukey-Kramer HSD in (D). Source data are available online for this figure.

hyperoxia therefore increases the severity of IAV infections by stimulating neutrophil recruitment.

## Ki-67 enhances neutrophil recruitment in mice exposed to PR8 influenza A virus or LPS

Neonatal hyperoxia increases the severity of Hkx31 (H3N2) infections because it increases the recruitment of neutrophils that are not normally recruited by this virus (Brandes et al, 2013; Tate

et al, 2011). In contrast, PR8 (H1N1) is a highly pathogenic strain of IAV because it causes significant neutrophil recruitment and neutrophil-mediated ALI. Since 20% of AT1 cells in mice exposed to room air express Ki-67 (Fig. 1D), we wanted to know if they might be responsible for the high pathogenicity of PR8 virus. To answer this question, Mki67^WT, Mki67^WT/CreERT, and Mki67^CreERT hypomorph mice exposed to room air were infected with PR8 (H1N1) IAV (Fig. 9A). As expected, Mki67^WT mice infected with PR8 lost weight with 100% mortality observed by post-infection

day 14 (Fig. 9B,C). Their lungs contained numerous neutrophils on post-infection day 3 (Fig. 9D,E) and extensively injured Krt5+ and Krt8+ fibrotic alveoli (Fig. 9F,G). In contrast, infected *Mki67^CreERT* hypomorph mice lost significantly less weight, and remarkably, none of the mice died. Furthermore, neutrophil recruitment was blunted in infected *Mki67^CreERT* mice compared to wild-type mice (Fig. 9D,E). Lungs of infected *Mki67^CreERT* mice also showed minimal alveolar Krt5 and Krt8 staining, and better alveolar structure than infected *Mki67^WT* mice (Fig. 9E–G). Intriguingly, *Mki67^WT/CreERT* mice displayed an intermediate response to infection when compared to *Mki67^WT* and *Mki67^CreERT* mice. We observed 50% survival by post-infection day 14, a modest recruitment of neutrophils, and the presence of patchy Krt5 and Krt8 staining in lungs that also contained regions of normal alveolar structure (Fig. 9A–G).

This finding prompted us to challenge *Mki67^CreERT* mice with lipopolysaccharide (LPS) to test whether Ki-67 was specifically modulating the response to IAV or is a general regulator of neutrophil recruitment. *Mki67^WT* and *Mki67^CreERT* mice were instilled intratracheally with LPS or phosphate-buffered saline (PBS) vehicle used as control (Fig. EV4). Leukocytes were quantified and identified in the lavage fluid collected three days later. LPS significantly increased the total number of immune cells, and this was attributed to a large increase in neutrophils. This increase was significantly dampened in *Mki67^CreERT* hypomorph mice exposed to LPS and was not seen in *Mki67^WT* or *Mki67^CreERT* hypomorph mice challenged with PBS. Based upon these findings, we conclude Ki-67 is a general enhancer of neutrophil recruitment in the lung.

### Ki-67 enhances basal and IL-1β-induced expression of *Cxcl* genes in mouse AT1-like cells

AT1 cells are historically difficult to isolate and culture. To further understand how Ki-67 affects Cxcl gene expression in AT1 cells, we created a novel AT1-like cell line by isolating AT2 cells from the Immortomouse (Jat et al, 1991), and allowing them to differentiate into AT1-like cells under conditions permissive for SV40 expression. One clonal line called AT1.1 was expanded for further study because it expresses the AT1-specific genes *T1α*, *Aqp5*, *Hopx*, *Igfbp2*, and *Gramd2*, but not the AT2-specific genes *Sftpa*, *Sftpb*, *Sftpc*, or *Sftpd* (Fig. EV5). Intriguingly, these cells lost the requirement for SV40 expression because they continued to proliferate and express high levels of Ki-67 even under non-permissive conditions.

These cells were then used to test the hypothesis that Ki-67 modifies the intensity of pro-inflammatory signals that stimulate NF-κB-dependent transcription of *Cxcl* genes and neutrophil recruitment during cytokine storms (Chan et al, 2021; Korbecki et al, 2022). AT1.1 cells were transfected with Ki-67 siRNA or scrambled control oligonucleotides. Silencing Ki-67 expression strongly reduced Ki-67 protein and mRNA expression (Fig. 10A,D). It did not affect the growth of the cells (Fig. 10B). However, silencing Ki-67 expression reduced basal and IL-1β-mediated stimulation of *RelA/p65*, *Cxlc1*, *Cxcl5*, and *IL-6*. It also blocked IL-1β stimulation of an NF-κB-dependent luciferase reporter (Fig. 10C), supporting its ability to suppress IL-1β stimulation of *RelA/p65*. The AT1.1 cells were then transfected with RelA/p65 siRNA to confirm that IL-1β activates NF-kB, which stimulates transcription of *Cxcl* genes. As expected, silencing RelA/p65 reduced basal and IL-1β stimulation of *Cxcl1*, *Cxcl5*, and *IL-6*.

However, it did not affect expression of *Ki-67* (Fig. 10D). Thus, Ki-67 regulates the intensity of IL-1β signaling that promotes NF-κB-dependent transcription of neutrophil chemoattractant molecules in AT1-like epithelial cells.

## Discussion

Neutrophils play an important role in defending the lung against infections, but their numbers must be tightly regulated to prevent excessive tissue injury and mortality. Why neutrophils are appropriately recruited in some individuals and not in others is poorly understood. We previously showed how neonatal hyperoxia enhances the severity of IAV infections in adult mice (Giannandrea et al, 2012; O'Reilly et al, 2008; Yee et al, 2017). Using this model, we discovered that neonatal hyperoxia increases the number of adult AT1 cells expressing Ki-67, an established proliferation marker used to identify aggressive tumor cells because it is highly expressed throughout the cell cycle but not in long-term quiescent cells (Sobecki et al, 2017). However, emerging evidence shows that Ki-67 is not sufficient or necessary for cell proliferation (Cidado et al, 2016; Cuylen et al, 2016; Sobecki et al, 2016). Instead, it re-localizes nucleolar material during mitosis, organizes peri-nucleolar heterochromatin, and regulates all phases of cancer development by controlling global gene expression (Andres-Sanchez et al, 2022). Taken together, it appears that Ki-67 modulates the intensity of gene expression that help cancer cells adapt to their environment. Consistent with this concept, we found that AT1 cells expressing Ki-67 are more adept at responding to environmental insults that stimulate inflammation and particularly pro-inflammatory signaling that promotes neutrophil recruitment. Neutrophil recruitment was attenuated in mice deficient in Ki-67 and enhanced when Ki-67 was overexpressed in the lung. Our studies using neonatal hyperoxia as a susceptibility factor suggest the severity of respiratory illness may be caused by early life environmental factors that influence the expression of Ki-67.

It was puzzling to find that adult AT1 cells express Ki-67 because they have historically been considered terminally quiescent and incapable of proliferation (Kauffman, 1980). This prompted us to consider whether AT2 cells might be the source of Ki-67+ AT1 cells. AT2 cells serve as bi-potential progenitor cells capable of self-renewal and differentiation into AT1 cells (Barkauskas et al, 2013). However, this appears to be restricted to adult AT2 cells because neonatal AT2 cells do not produce adult AT1 cells during postnatal growth and development (Penkala et al, 2021; Yee et al, 2016). Neonatal hyperoxia may activate an injury program normally seen in the adult because it stimulates proliferation of AT2 cells that are slowly depleted when they differentiate into an AT1 cell (Yee et al, 2014). Genetic lineage-mapping experiments in the current study found that neonatal hyperoxia drives AT2 cells to differentiate into AT1 cells and that they also express Ki-67, implying that these AT1 cells retained some molecular memory of proliferating as an AT2 cell during hyperoxia.

Ki-67 was also detected in a small number of AT2 cells, but their numbers were not different between mice exposed to room air or hyperoxia as neonates. These Ki-67+ AT2 cells may have retained expression of Ki-67 after they exited the cell cycle during fetal or postnatal lung development, perhaps reflecting a small tetraploid population seen after acute lung injury (Weng et al, 2022).

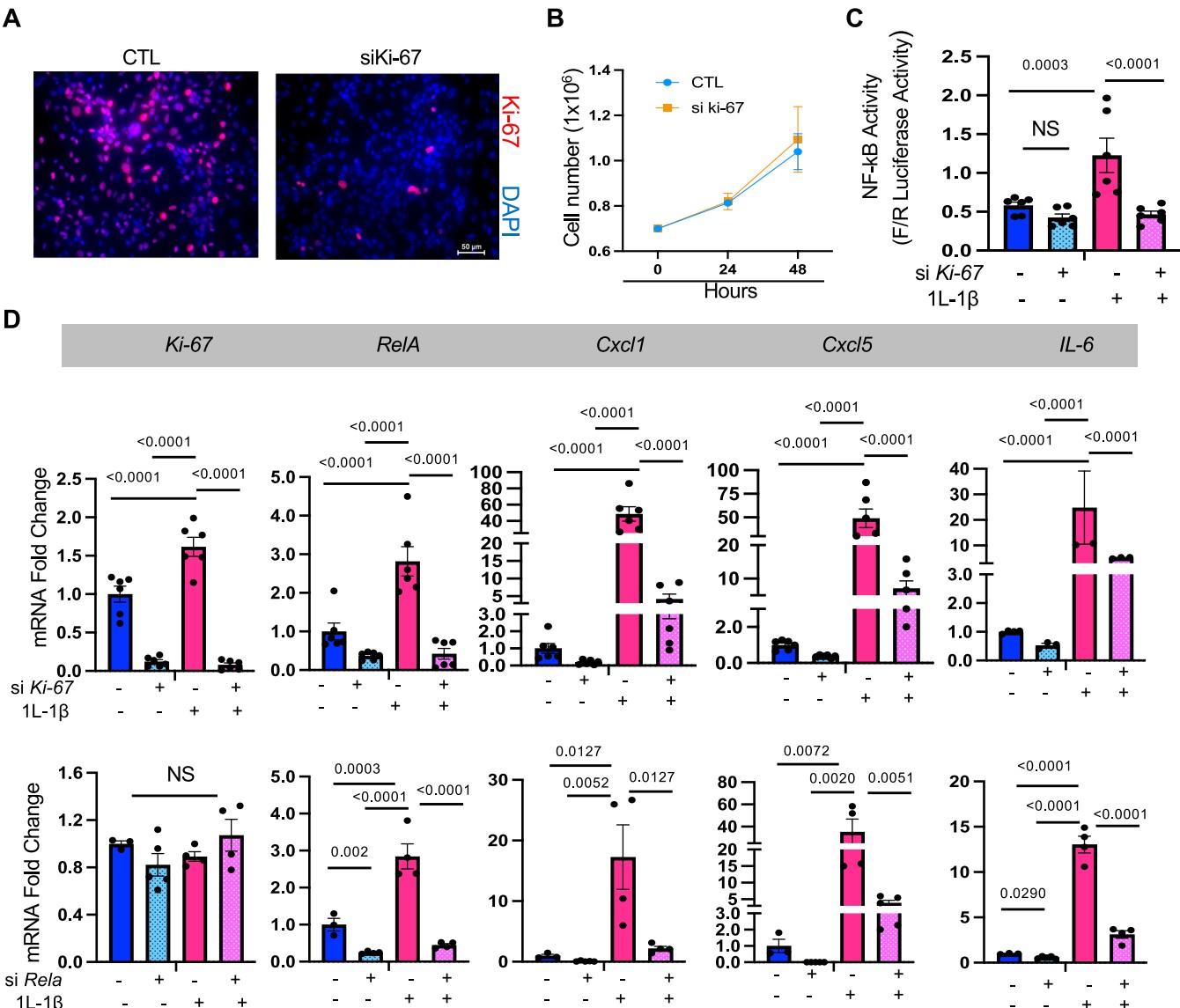

**Figure 10. Ki-67 enhances RelA-dependent transcription of Cxcl genes in mouse AT1.1 epithelial cells.**

(A) AT1.1 cells transfected with Ki-67 siRNA for 48 h were fixed and stained for Ki-67 (magenta) and counterstained with DAPI (blue). Scale bar = 50 μm. (B) AT1.1 cells were transfected with Ki-67 siRNA or control oligonucleotides. The equal number of cells were plated on day 0 and counted 24 and 48 h later. Data is graphed as a mean number of cells ± standard deviation. $n = 3$ cultures per group. (C) AT1.1 cells were transfected with Ki-67 or a scrambled control for 48 h. The cells were then transfected with NF-κB-luciferase reporter and cultured for an additional 24 h in control media or media containing 0.5 ng/ml IL-1β. Luciferase activity was then graphed as mean fold change ± standard deviation relative to control cells cultured in the absence of IL-1β. $n = 6$ cultures per group. (IL-1β vs control: $P = 0.0003$; IL-1β vs IL-1β and siKi67: $P < 0.0001$). (D) AT1.1 cells were transfected with Ki-67 siRNA or scrambled controls for 48 h and then cultured for 24 h in 0.5 ng/ml IL-1β or control media. QRT-PCR was used to detect expression of Ki-67, RelA, Cxcl1, Cxcl5, and IL-6. Data are graphed as mean fold change ± standard deviation compared to control cells. $n = 6$ for assessing siKi67 on Ki-67, RelA, Cxcl1, and Cxlc5 expression and $n = 4$ for all other measurements. (Ki-67, RelA, Cxcl1, Cxcl5, and IL-6: Control vs siKi67 cells: $P < 0.0001$; SiKi67 vs IL-1β cells: $P < 0.0001$; IL-1β vs IL-1β and siKi67 cells: $P < 0.0001$). (RelA: Control vs siRelA cells: $P < 0.002$; Control vs IL-1β cells: $P = 0.0003$; siRelA vs Il-1β cells: $P < 0.0001$; Il-1β vs siRelA and IL-1β: $P < 0.0001$). (Cxcl1: Control vs siRelA cells: $P < 0.0052$; Control vs IL-1β cells: $P = 0.0127$; siRelA vs Il-1β cells: $P < 0.0001$; Il-1β vs siRelA and IL-1β: $P < 0.0001$). (Cxcl5: Control vs IL-1β cells: $P = 0.0072$; siRelA vs Il-1β cells: $P = 0.0020$; Il-1β vs siRelA and IL-1β: $P = 0.0051$). (IL-6: Control vs siRelA cells: $P = 0.0290$; Control vs IL-1β cells: $P < 0.0001$; siRelA vs Il-1β cells: $P < 0.0001$; Il-1β vs siRelA and IL-1β: $P < 0.0001$). Data in (B–D) graphed as mean ± standard deviation. Data reflect biological replicates analyzed by one-way ANOVA using Tukey-Kramer HSD (B–D). Source data are available online for this figure.

Alternatively, they may reflect a small population of AT2 cells that derive from a HOPX+ AT1 lineage controlled by HIPPO signaling (Penkala et al, 2021). We were unable to test this possibility using the Aqp5$^{Cre}$; R26$^{mTmG}$ mice because the Aqp5$^{Cre}$ mouse targets some AT2 cells and in a strain-specific manner (Flodby et al, 2010). Although we could detect EGFP in some SFTPC+ or ABCA3+

AT2 cells by immunohistochemistry, adding antibodies against Ki-67 generated high background staining that limited our ability to identify Ki-67 in lineage-labeled AT1 cells that became an AT2 cell. Experiments using the Hopx$^{CreERT}$ or the new Gramd2$^{CreERT}$ mice may help define the source of Ki-67 expression in AT2 cells (Penkala et al, 2021; Yang et al, 2023); however, it is important to

mention that none of these drivers are strictly specific for AT1 cells (personal observation and communication with Zea Borok). We recommend that multiple AT1-Cre lines be used to rigorously determine whether the AT1 lineage can generate Ki-67-positive AT2 cells. In summary, we conclude that Ki-67 is primarily expressed by AT1 and AT2 cells in the adult lung, because it is rarely detected in bronchioles and mainstem bronchi by immuno-histochemistry or in CD31+ or CD144+ endothelial cells by flow cytometry.

While attempting to lineage map Ki-67+ AT1 cells, we unexpectedly discovered that *Mki67^CreERT* mice are hypomorphic for Ki-67 expression in the lung. Despite the profound loss of Ki-67, these mice developed histologically normal lungs that still became simplified when exposed to neonatal hyperoxia. It may be that Ki-67 is not required for lung development because another Ki-67 hypomorph with a mutant translation initiation site also produces viable mice (Sobecki et al, 2016). However, it cannot be ruled out that a low level of Ki-67 sufficient to support lung development is retained in these hypomorphic models. Consistent with this idea, Ki-67 can be detected in proliferating cells of *Mki67^CreERT* hypomorph mice such as during lung development. However, the number of Ki-67+ cells is significantly less than seen in wild-type control even though both mice develop normal healthy lungs. Careful examination of Ki-67 staining failed to detect changes in staining intensity that might distinguish Ki-67 as a proliferation marker versus a regulator of inflammatory gene expression. Thus, it remains to be determined whether levels of Ki-67 within the cell impart different biological functions.

Our findings however provide clear evidence that Ki-67 expressed by non-proliferating alveolar epithelial cells enhances the expression of pro-inflammatory genes driving neutrophil recruitment when mice are infected with IAV. This helps explain why neonatal hyperoxia increases the severity of Hkx31 (H3N2) infections, but marginally increases the severity of PR8 (H1N1) infections (Domm et al, 2017; O'Reilly et al, 2008). Hkx31 is a low-pathogenic strain of IAV that primarily infects airway epithelial cells and macrophages with minimal recruitment of neutrophils. Neonatal hyperoxia increases the number of Ki-67+ AT1 cells, resulting in high expression of chemokines that recruit neutrophils when mice are infected with Hkx31. Neutrophils kill epithelial cells via oxidation and NETosis when they are recruited to localized sites of inflammation and infection (Castanheira and Kubes, 2019; Knaapen et al, 1999). In contrast, PR8 is a highly pathogenic virus because neutrophils are already excessively recruited by this virus. Neonatal hyperoxia increases the number of AT1 cells expressing Ki-67, but this is not sufficient to increase the severity of PR8 infections, perhaps because neutrophil recruitment is already at a maximum. Thus, a small number of Ki-67+ cells is sufficient to drive ALI with a high pathogen virus like PR8, while a larger number of Ki-67+ cells drive ALI during infection with a low pathogen virus such as Hkx31.

The effect of Ki-67 on neutrophil recruitment is not specific for viral infections. Our finding that Ki-67 drives neutrophil recruitment in mice challenged with LPS, an immunogenic component of bacterial cell walls, indicates it may play a critical role in pneumonia. It may also regulate neutrophil influx and acute lung injury caused by bleomycin. We previously showed that neonatal hyperoxia increases bleomycin-induced lung fibrosis attributed to an early increase in neutrophils and increased TGF-β activity (Yee et al, 2013). Depleting neutrophils using anti-Gr-1 antibody

reduced TGF-β activity and the extent of fibrosis in mice exposed to bleomycin. At that time, we did not consider that these observations would explain why neonatal hyperoxia increases the severity of IAV infections, because neutrophils are poorly recruited by Hkx31 virus. The collective findings suggest Ki-67 expressed by alveolar epithelial cells is a general modulator of neutrophil recruitment that drives the severity of acute lung injury.

Our findings in mice have broad translational implications. Since Ki-67 is expressed by proliferating cells, it may play a role in regulating the inflammatory response during tissue repair. For example, Ki-67 expressed by proliferating airway (basal, Club) or alveolar (AT2) cells may modulate the intensity of genes regulating the interaction between epithelial progenitors and immune cells. Hypothetically, loss of Ki-67 as proliferating cells exit the cell cycle would dampen epithelial inflammatory signaling and potentially resolve inflammation. Conversely, Ki-67 would continue to be expressed when repair is not effective and thus potentially drive persistent inflammation. Ki-67 expressed by proliferating fibroblasts might drive inflammation and fibrotic lung disease. High levels of Ki-67 present in proliferating cells may also drive inflammation and the severity of respiratory infections in children, whose lungs are still growing and are susceptible to RSV-mediated respiratory illness. Consistent with this hypothesis, instilling keratinocyte growth factor (KGF) into lungs of adult mice drives proliferation of normally quiescent AT2 cells. It also increases the severity of IAV infection (Nikolaidis et al, 2017). High levels of Ki-67 seen in non-small cell lung adenocarcinomas associate with high levels of CXCL chemokines and the presence of tumor-associated neutrophils (Masucci et al, 2019; Zhou et al, 2023). High expression of Ki-67 by malignant epithelial cells may shape how neutrophils are recruited to the tumor microenvironment. Consistent with this hypothesis, neutrophils are poorly recruited to mammary tumors lacking Ki-67 orthologously transplanted into a wild-type mouse compared to mammary tumors that express Ki-67 (Mrouj et al, 2021). Finally, it is important to mention that our findings may reflect properties of Ki-67 to modulate the intensity of inflammatory gene expression in any cell and organ. Understanding how Ki-67 functions in different types of cells and conditions is important because it could change its use from a clinical marker of proliferation to a therapeutic target.

Transcriptomic studies in cancer cells indicate that Ki-67 plays an important role in modulating the intensity of gene expression through its ability to bind and organize heterochromatin and thus shape chromatin landscapes (Sobecki et al, 2016). In this capacity, the carboxy-terminal chromatin binding domain of Ki-67 helps maintain DNA integrity during mitosis (Garwain et al, 2021). Acute depletion of Ki-67 has been seen to create DNA damage during mitosis that activates p53-dependent checkpoints. We found Ki-67 also enhances the expression of TNF-α, IL-1β, and members of the Cxcl family of chemokines when mice are infected with IAV. These genes all contain canonical NF-kB binding sites in their promoters (Korbecki et al, 2022). Viral and bacterial infections increase expression of TNF-α, IL-1β, and other cytokines that stimulate NF-kB activation responsible for amplifying and driving expression of downstream cytokines/chemokines such as CXCL5. Our studies with AT1.1 cells revealed that Ki-67 shapes the intensity of IL-1β activation of NF-κB signaling and thus transcription of several *Cxcl* genes responsible for neutrophil recruitment. Inappropriate NF-kB inflammatory signaling is often an underlying feature of acute lung injury. TNF-α and IL-1β can activate NF-kB canonical signaling through phosphorylating IkBa, a protein that sequesters RelA/p65 in the cytoplasm. NF-kB can be

activated via other signaling molecules, including Protein Kinase C (PKC) and p38 MAP kinase. p38 can be activated by p21-activated kinases (PAKs) and Cell Division Cycle 42 (Cdc42) homolog. A PAK1-interacting protein (PAK1IP1) can disrupt PAK1 activation of p38. Interestingly, PAK1IP1 and Cdc42 are two of 406 RNA-binding proteins that interact with Ki-67 in nuclear pulldown assays (Sobecki et al, 2016). While the interaction of Ki-67 with PAK1IP1 and Cdc42 supports the current study showing Ki-67 modulates IL-1β signaling, how it does so in AT1 cells remains to be determined.

It is important to point out some limitations in our study. Genetic lineage labeling studies using *Sftpc^CreERT* mice revealed that Ki-67+ AT1 cells were derived from AT2 cells that may have proliferated during hyperoxia. However, it is possible that AT1 cells expressing Ki-67 do not entirely derive from AT2 cells because neonatal hyperoxia can increase AT1 cell expansion when Hippo signaling is disrupted in Hopx+ cells. (Penkala et al, 2021). Although Ki-67 was primarily detected in CD326+ epithelial cells, it could be expressed by fibroblasts, pericytes or other types of lung cells that we did not study. Regardless of the cellular source of expression, neutrophil recruitment was attenuated in *Ki-67* hypomorphs and restored when Ki-67 was overexpressed in the lung. This provides the strongest evidence that resident cells of the lung expressing Ki-67 drive inflammation during infections. But we cannot formally rule out a role for Ki-67 in producing neutrophils in bone marrow (granulopoiesis) or influencing their ability to function during infection. In fact, the process of NETosis is tightly integrated with the cell cycle, suggesting that Ki-67 may also be important for this aspect of neutrophil biology (Amulic et al, 2017). Finally, our studies did not discern whether the effects of Ki-67 on inflammatory gene expression are restricted to a specific type of cell in the lung. The plasmid over-expression studies suggest that cell specificity may not matter because neutrophil recruitment was enhanced when Ki-67 was randomly overexpressed in the lung. However, it will be important to sort this out because the experiments may show that Ki-67 affects other transcriptional networks in a cell-specific manner.

In summary, we discovered a previously unappreciated mechanism by which the oxygen environment experienced at one point in life affects the severity of respiratory viral infections later in life via sustained expression of Ki-67. Although Ki-67 is widely known as a marker of cell proliferation, we found that it enhances the production of inflammatory cytokines and chemokines responsible for recruiting neutrophils into the lung. High levels of oxygen at birth increase the expression of Ki-67 in adult mice and thus the severity of infection-related illness. The severity of respiratory infections therefore may be modulated by mitogens and other environmental factors, such as hyperoxia that increase expression of Ki-67.

# Methods

### Reagents and tools table

| Reagent/resource | Reference or source | Identifier or catalog number |
| --- | --- | --- |
| **Experimental models** | | |
| C57BL/6J (*M. musculus*) | Jackson Lab | RRID:IMSR_JAX:000664 |
| Sftpc^EGFP (*M. musculus*) | Lo et al, 2008 | N/A |
| Aqp5^Cre (*M. musculus*) | Flodby et al, 2010 | N/A |
| Mki67^CreERT (*M. musculus*) | Jackson Lab | RRID:IMSR_JAX:0229803 |
| Sftpc^CreERT (*M. musculus*) | Jackson Lab | RRID:IMSR_JAX:028054 |
| R26^mTmG (*M. musculus*) | Jackson Lab | RRID:IMSR_JAX:007676 |
| H-2Kb^tsA58 (*M. musculus*) | Jackson lab | RRID:IMSR_JAX:032619 |
| AT1.1 cells (*M. musculus*) | This study | N/A |
| Hkx31 (H3N2) influenza A virus | O'Reilly et al, 2008 | N/A |
| PR8 (H1N1) influenza A virus | Domm et al, 2017 | N/A |
| **Recombinant DNA** | | |
| pcDNA3.1 | ThermoFisher | V79020 |
| pcDNA3.1-Ki-67mCherry plasmid | This study | "Methods" |
| **Antibodies** | | |
| Rat anti-CD31 | BD Biosciences | Cat #740356 |
| Rat anti-CD34 | BD Biosciences | Cat #751621 |
| Rat anti-CD45 | BD Biosciences | Cat #559864 |
| Rat anti-CD144 | BioLegend | Cat #138106 |
| Rat anti-CD326 | BD Biosciences | Cat #563214 |
| Mouse anti-Hopx | Santa Cruz | Cat #398703 |
| Mouse anti-Ki-67 | BioLegend | Cat #6525426 |
| Rat anti-CD16/CD32 | BD Biosciences | Cat # 553141 |
| Mouse anti-Cytokeratin 5 | Invitrogen | MA5 -14473 |
| Rat anti-Cytokeratin 8 | DSHB | TROMA-1 |
| CXCL5 | Invitrogen | PA5-115069 |
| Goat anti-EGFP | Abcam | Ab6662 |
| Rabbit anti-Ki-67 | Abcam | AB15580 |
| Hamster anti-T1α/Podoplanin | DSHB | 8.1.1 |
| Rabbit anti-Phospho-Histone H3 (Ser10) | ThermoFisher | PA5-17869 |
| Rabbit anti-SFTPC | Seven Hill Bioreagents | WRAB-9337 |
| Rabbit anti-Cit-histone H3 | Abcam | AB5103 |
| Rat anti-BrdU | Abcam | Ab6326 |
| Rat anti-Ly6G (1A8) | BioX Cell | BE0075-1 |

| Reagent/resource | Reference or source | Identifier or catalog number |
|---|---|---|
| **Oligonucleotides and other sequence-based reagents** | | |
| Aqp5_forward primer | NM_009701.4 | AGATCTCCATAGCCTTTGGCCT |
| Aqp5_reverse primer | NM_009701.4 | AGCAGAGAGATCTGGTTGCCTA |
| Cxcl1_forward primer | NM_008176.3 | GCTTGAAGGTGTTGCCCTCAG |
| Cxcl1-reverse primer | NM_008176.3 | AAGCCTCGCGACCATTCTTG |
| Cxcl2_forward primer | NM_009140.2 | GCGCTGTCAATGCCTGAAGA |
| Cxcl2_reverse primer | NM_009140.2 | TTTGACCGCCCTTGAGAGTG |
| Cxcl4_forward primer | NM_019932.5 | GTTCCCCAGCTCATAGCCACC |
| Cxcl4_reverse primer | NM_019932.5 | TTATATAGGGGTGCTTGCCGGT |
| Cxcl5_forward primer | NM_009141.3 | CAGTGCCCTACGGTGGAAG |
| Cxcl5_reverse primer | NM_009141.3 | TAGCTTTCTTTTTGTCACTGCCC |
| Gramd2_forward primer | NM_001033498.2 | GGACACATTTCCCTCTAGCAAC |
| Gramd2_reverse primer | NM_001033498.2 | ATTTGCTCCGTAGTGTCCCT |
| Hopx_forward primer | NM_001159901.1 | CGGGCCATCTGGTTCCC |
| Hopx_reverse primer | NM_001159901.1 | GCTGCTTAAACCATTTCTGCG |
| Igfbp2_forward primer | NM_008342.4 | AAGCATGCGGCGTCTACAT |
| Igfbp2_reverse primer | NM_008342.4 | TCGTCATCACTGTCTGCAACC |
| 1L-1β_forward primer | NM_008361.4 | TGCCACCTTTTGACAGTGATG |
| 1L-1β_reverse primer | NM_008361.4 | TGATGTGCTGCTGCGAGATT |
| mKi-67_forward primer | NM_001081117.2 | CCTGCCTCAGATGGCTCAAA |
| mKi-67_reverse primer | NM_001081117.2 | GGTTCCCTGTAACTGCTCCC |
| Rela_forward primer | NM_009045.4 | CCTCTGGCGAATGGC |
| Rela_reverse primer | NM_009045.4 | GAGGGGAAACAGATCGTCCA |
| Sftpa_forward primer | NM_023134.4 | TTCCAGGGTTTCCAGCTTACCT |
| Sftpa_reverse primer | NM_023134.4 | AGTTGACTGACTGCCCATTGGT |
| Sftpb_forward primer | NM_147779.1 | TGGAACACCAGTGAACAGGCTA |
| Sftpb_reverse primer | NM_147779.1 | GCATGTGCTGTTCCACAAACTG |
| Sftpc_forward primer | NM_011359.2 | TGATGGAGAGTCCACCGGATTA |
| Sftpc_reverse primer | NM_011359.2 | CCTACAATCACCACGACAACGA |
| Sftpd_forward primer | NM_009160.2 | CTGATGGCCGAAGTGTTGGA |
| Sftpd_reverse primer | NM_009160.2 | CAGTAGCAGAACGTGGGGAG |
| T1α_forward primer | NM_010329.2 | AGCAAAGCCAAGACAGTATCGC |
| T1α_reverse primer | NM_010329.2 | TTAGGACTGGGCTGGAATGTGT |
| Tnf-α_forward primer | NM_013693.3 | GTCCCCAAAGGGATGAGAAGT |
| Tnf-α_reverse primer | NM_013693.3 | TTTGCTACGACGTGGGCTAC |
| 18S RNA_forward primer | NR_003278.1 | CGGCTACCACATCCAAGGAA |
| 18S RNA_reverse primer | NR_003278.1 | GCTGGAATTACCGCGGCT |
| iTaq universal SYBR | Bio-Rad | Cat # 1725124 |
| iScript™ cDNA Synthesis Kit | Bio-Rad | CAT# 1708891 |
| **Chemicals, enzymes, and other reagents** | | |
| Tamoxifen | Sigma-Aldrich | Cat #T5648 |
| Oxygen | http://www.airgas.com | Cat #OX USP180LT230 |
| Lipopolysaccharide | Sigma-Aldrich | Cat #L9143 |
| 5-bromo-2′-deoxyuridine | Life Technologies | Cat #000103 |
| Dip-Stain kit | Volu-SOL | Cat #VDS-100 |
| Gomori's Trichrome stain kit | ThermoFisher Scientific | Cat #87020 |
| TUNEL assay kit | EMD Millipore | Cat #S7101 |
| 4′,6-diamidino-2-phenylindole (DAPI) | Southern Biotech | Cat #0100-20 |
| RBC lysis buffer | BioLegend | Cat #420301 |
| Live/Dead stain kit | Invitrogen | Cat #L34957 |
| Intracellular Fixation & Permeabilization Buffer Set | Invitrogen | Cat # 88-8824-00 |
| DMSO | Sigma-Aldrich | Cat # D8418 |
| Corning Dispase | Sigma-Aldrich | Cat # CLS354235 |
| Low-melt agarose | Sigma-Aldrich | Cat # A9414 |
| IL-1β/IL-1F2 | R&D Systems | Cat #401-ML-010/CF |
| IFN-γ | R&D Systems | Cat #485-ML |
| TRIzol™ Plus RNA Purification Kit | Invitrogen | Cat #12183555 |
| RNeasy Plus Mini Kit | Qiagen | Cat #74134 |
| Cell culture chamber slide system | Nest Scientific USA | Cat #230102 |
| Renilla Luciferase Assay System | Promega | Cat #E2810 |

| Reagent/resource | Reference or source | Identifier or catalog number |
|---|---|---|
| **Software** | | |
| FlowJo v10 | FlowJo LLC | N/A |
| JMP 16 statistical software | SAS Institute | N/A |
| Prism | GraphPad | N/A |
| **Other** | | |
| ECM830 electroporator | BTX, Harvard Apparatus | Cat #45-0662 |
| BD LSR II | BD Biosciences | Cat #339101 |
| BD FACS Aria III | BD Biosciences | Cat #3374327 |
| Nikon E-800 fluorescence microscope | Nikon | Eclipse E-800 |
| SPOT-RT slidercamera | Diagnostic Instruments | RT-230 |
| CFX96TM PCR machine | Bio-Rad | Cat #1845096 |
| CFX384TM PCR machine | Bio-Rad | Cat #12011319 |
| Shandon Cytospin 3 Centrifuge | Thermo | Cat #59900102 |

## Methods and protocols

### Exposing newborn mice to hyperoxia

C57BL/6J (RRID:IMSR_JAX:000664), *Mki67*CreERT (RRID:IMSR_JAX:0229803), *R26*mTmG (RRID:IMSR_JAX:007676), and *Sftpc*CreERT (RRID:IMSR_JAX:028054) mice were purchased from the Jackson Laboratories (Bar Harbor, ME). *Sftpc*EGFP mice (Lo et al, 2008) were obtained from Dr. Brigid Hogan (Duke University), and *Aqp*Cre mice were obtained from Dr. Zea Borok (University of California at San Diego) (Flodby et al, 2010). Mice were genotyped by PCR using DNA isolated from tail snips and gene-specific primers for *Mki67*CreERT, *R26*mTmG, and *Sftpc*CreERT provided by the Jackson Laboratories or as described in publications for *Sftpc*EGFP mice (Lo et al, 2008) and *Aqp*Cre (Flodby et al, 2010) mice. Genetic lineage labeling studies were initiated by crossing Ki-67CreERT, *Sftpc*CreERT or *Aqp5*Cre with *R26*mTmG mice. Enhanced green fluorescent protein (EGFP) was activated in progeny of CreERT crosses by injecting pregnant dames with one dose tamoxifen (0.05 mg/g) on E18.5 or injecting adults with 4 daily sequentially doses (0.25 mg/g) tamoxifen (Sigma-Aldrich, St. Louis, MO, T5648) once a day for 4 days or corn oil vehicle as control.

Newborn mice were exposed to room air or hyperoxia (100% oxygen) between postnatal days (PND) 0 and 4 (Yee et al, 2014). Dams were rotated between room air and hyperoxia every 24 h to reduce oxygen toxicity to their lungs. Oxygen levels, humidity of 40–70%, and temperature of 37 °C were monitored at least twice per day. Mice exposed to hyperoxia were then returned to room air with their control siblings until they reached 8–12 weeks of age. All mice were housed in a specific pathogen-free environment and provided food and water *ad libitum* according to a protocol (2007-121E) approved by the University Committee on Animal Resources. All experiments performed with mice were in accordance with the relevant guidelines and regulations of this committee.

### Infecting mice with the influenza A virus or LPS

Adult mice (8–12 weeks of age) were anesthetized with avertin and then infected intranasally with 120 hemagglutinating units of HKx31 (H3N2) or $1 \times 10^3$ plaque-forming units of PR8 (H1N1) strains of IAV in phosphate-buffered saline (Domm et al, 2017; O'Reilly et al, 2008). Additional mice were exposed to 5 mg/kg lipopolysaccharide (Sigma-Aldrich) delivered intranasally. Mice were randomly chosen and administered virus or LPS on an alternating basis with mice given saline vehicle used as a control. Mice were then monitored as they recovered from anesthesia. Neutrophils were depleted by injecting mice intraperitoneally immediately prior to infection with (300 μg) anti-Ly6G (1A8) or a non-specific IgG antibody (BioX Cell, Lebanon, New Hampshire). Proliferating cells were labeled with 1% (vol/wt) 5-bromo-2′-deoxyuridine (BrdU) injected into mice 1% (vol/wt) 5-bromo-2′-deoxyuridine (BrdU) 2 h prior to sacrifice (Life Technologies, Carlsbad, CA) (Domm et al, 2017).

### Electroporation-mediated gene delivery

The Ki-67 expression plasmid was generated by Gateway recombination from the entry vector pENTR-3C, containing full-length ORF cloned from human cDNA (Sobecki et al, 2016), into the destination vector pmCherry-N1-RfC. Correct recombination was determined by sequencing the plasmid using nanopore sequencing. The Ki-67 open-reading-frame fused with a C-terminal mCherry sequence was then cloned into BamH1 and Not1 sites of the pcDNA3.1 expression plasmid. The Ki-67mCherry and parent pcDNA3.1 plasmids were purified with Qiagen Giga-prep kit and suspended at 1 mg/ml in 10 mmol/L Tris, pH 8.0, 1 mmol/L EDTA, and 140 mmol/L NaCl. Adult mice were lightly anesthetized with isoflurane and placed in a supine position for attaching pediatric cutaneous pacemaker electrodes on both sides of the chest. Plasmid (50 μl) were administered to the lungs by tracheal instillation during inspiration, followed by a series of eight consecutive square-wave electric pulses (220 V/cm for 10 ms each) using an ECM830 electroporator (Gentronics, West Hollywood, CA) (Zhou et al, 2008). Mice were monitored as they recovered from anesthesia and returned to the vivarium until further study.

### Bronchoalveolar lavage (BAL) cell numbers

The trachea was exposed on anesthetized mice and an 18 g catheter inserted into a small slit. Lungs were then lavaged three times with 1 ml of ice-cold phosphate-buffered saline. The bronchoalveolar lavage was pooled, centrifuged at $600 \times g$, and the supernatant stored at −80 °C. The cell pellet was resuspended in Dip-Stain Kit (Volu-SOL, VDS-100, Salt Lake City, UT). Cells were counted with a hemacytometer, and $1 \times 10^5$ cells were cytospun onto microscope slides that were stained with Dip-Stain solution (Volu-SOL, VDS-100 Sal for visual identification and counting of leukocytes. Approximately 500 cells were counted on each slide.

### Lung histology and staining

Lungs were inflation fixed in 10% neutral-buffered formalin at 25 cm water pressure for 20 min. Lungs were removed from the thorax and further fixed in buffer overnight at 4 °C. Lungs were then dehydrated in graded steps of 50% and 70% ethanol for 30 min and then storage in 100% ethanol. The dehydrated lungs were embedded in paraffin and sectioned at 5 μm. The sections were deparaffinized in 2 × 5 min xylene and rehydrated with 2 × 5 min of

100% ethanol, 3 min of 95% ethanol, 3 min of 70% ethanol, and 5 min of distilled deionized water before staining with hematoxylin and eosin, Gomori's trichrome (Thermo Scientific, catalog number 8709), a TUNEL assay (EMD Millipore, catalog number S7101) or with antibodies against specific proteins. For staining with antibodies or TUNEL assay, antigen retrieval was performed by incubating the section in 10 mM citrated buffer (pH 6.0) for 100 °C for 5 min and 50 °C for 15 min, then washed and blocked with blocking buffer (3% of normal serum) for 1 h at room temperature. Primary antibodies (1:100 or 200) were added to sections and incubated in a humidified staining box overnight at 4 °C. The stained sections were washed 3× in phosphate-buffered saline containing 1% Tween-20 and incubated with Alexa Fluor or Fluorescein (FITC)-conjugated Donkey secondary antibody (1:200; Jackson ImmunoResearch Laboratories, West Grove, PA) for 1 h in the dark at room temperature. Stained sections were then counterstained with 4′,6-diamidino-2-phenylindole (DAPI) before visualizing tissues with a Nikon E-800 fluorescence microscope (Nikon, Melville, NY). Images were captured with a SPOT-RT slider digital camera (Diagnostic Instruments, Sterling Heights, MI). Single- and double-positive cells were quantified from 4 to 6 random images (Yee et al, 2006).

### RT-PCR

Total RNA was isolated from the lung using TRIzol™Plus RNA purification kit (Invitrogen) following the manufacturer's instructions. cDNA was synthesized using the iScript cDNA synthesis kit (Bio-Rad Laboratories, Hercules, CA). Real-time quantitative PCR (qRT-PCR) used iTaq universal SYBR (Bio-Rad) and CFX Opus 96 Real-Time PCR System (Bio-Rad). Primers used for qRT-PCR are listed in the Reagents and Tools table. Gene expression was calculated relative to 18S within samples and expressed as fold change over average expression.

### Flow cytometry

To obtain single-cell suspensions from lung tissue, lung cells were collected by tracheal instillation of 2 ml Dispase (Gibco Life Technologies, Grand Island NY) followed by 0.45 ml of 1% low-melt agarose (Sigma, St. Louis, MO). The trachea was sutured closed, and the lung was immediately covered with ice to harden the agarose. The lung was then incubated in 2 ml Dispase at room temperature for 45 min and then finely minced with scissors in a solution containing type IV bovine pancreatic DNase (30 μg/mL; Sigma-Aldrich), HEPES buffer (Life Technologies), and 1% fetal bovine serum. Cells were collected by sequential filtering through 100 and then 75-μm cell strainers (Fisher Scientific, Waltham, MA). Red blood cells (RBC) were lysed with RBC lysis buffer in ice for 10 min (BioLegend, San Diego, CA), and then cells were transferred to staining buffer (Phosphate-buffered saline with 10% FBS). Cells were incubated with anti-mouse CD16/DC32 Fc block (BD Pharmingen) diluted 1:400 in staining buffer for 10 min at 4 °C. Cells were then stained with Live/Dead stain (Live/Dead fixable dead cell kit, Invitrogen) with aliquots of the sample exposed to 90 °C heat for 1 min serving as a positive control for dead cells. Approximately $1 \times 10^6$ cells were then incubated with antibodies used to identify different types of cells. Single stained channels were used for compensation and fluorophore minus one (FMO) as controls used to define gating parameters for analysis. A BD Cytofix/Cytoperm™ Intracellular Fixation and permeabilization

buffer set kit (Invitrogen) was used when cells were stained with an antibody to Ki-67. Flow cytometry was performed using BD LSR II and BD FACS Aria III flow cytometers (BD Biosciences), and data were analyzed with FlowJo software (BD Biosciences).

### Generation of AT1.1 cells, reporter gene transfection, and luciferase assay

Sftpc$^{EGFP}$ mice were mated to H-2Kb$^{tsA58}$ (RRID:IMSR_JAX:032619) immortomice that express a temperature-sensitive allele of SV40 T antigen under control of the mouse major histocompatibility complex H-2Kb promoter (Jat et al, 1991) (Fig. EV5). EGFP+ AT2 cells were isolated from the mice and plated at $0.5 \times 10^6$ cells per 35 mm dish in Dulbecco's modified Eagle's medium supplemented with 100 units of penicillin and 100 U/ml recombinant mouse IFN-γ used to increase expression of the H-2kb promoter and at 33 °C permissive for SV40 expression. Cultures were refed with fresh media every 48–72 h. Two clones spontaneously expanded as EGFP expression was lost. The two clones were expanded under non-immortalizing conditions of 37 °C and in the absence of IFN-γ. Total RNA was isolated with Trizol Reagent (Invitrogen), 1 μg of RNA was synthesized to cDNA using iScript™ cDNA Synthesis Kit (Bio-Rad). β-actin was used as a loading control. AT1.1 expressed *T1* and *Aqp5,* while AT2.1 expressed *T1α* but not *Aqp5*. Consistent with the loss of AT2 phenotype, both cell lines did not express *Sftpa, Sftpb, Sftpc,* or *Nkx2.1* when compared to MLE15 cells, an SV40 immortalized line of mouse AT2 cells that also express AT1 genes, and the mouse lung. To get high purity of cells, the AT1.1 cells were sorted by staining with antibodies to CD326 and T1α followed by assessing gene expression using qRT-PCR. Total RNA was isolated using RNeasy Plus Mini Kit (Qiagen), then cDNA was synthesized was performed using iScript™ cDNA Synthesis Kit (Bio-Rad) with 1 μg of cDNA used to evaluate gene expression with 18S RNA as a loading control. The cells were not authenticated by STR profiling or tested for Mycoplasma. However, qRT-PCR revealed that they express high levels of the AT1-specific genes *T1α, Aqp5, Hopx, Igfbp2,* and *Gramd2,* but not the AT2-specific genes *Sftpa, Sftpb, Sftpc* or *Sftpd*. AT1.1 cells ($0.5 \times 10^6$ cells /35 mm) were cultured overnight at 37 °C to allow cells to adhere to the plates. The cells were transfected with 30 pM of pooled siRNA targeting *Ki-67* or *Rela* (Horizon Discovery, L-059904-00-0005 and L-040776-00-0005) using Lipofectamine RNAiMax (ThermoFisher Scientific, 13778075) and cultured for 48 h before culturing in 0.5 ng/ml of 1L-1β (R&D Systems, 401-ML-010/CF) for an additional 24 h. RNA was then isolated, or cells were fixed with 4% of paraformaldehyde for 10 min, washed, then permeabilized with 0.1% Triton X-100 for 10 min at room temperature. Cells were incubated with anti-Ki-67 antibody overnight at 4 °C, then washed and incubated with Alexa Fluor-conjugated Donkey secondary antibody (1:200; Jackson ImmunoResearch Laboratories, West Grove, PA) for 1 h in the dark room temperature as described for staining lungs.

To assess NF-κB activity, cells were transfected with an NF-κB-luciferase expression plasmid using DEAE-Dextran as described (Bijli et al, 2012). NF-κB-luciferase reporter plasmid contains five copies of the consensus NF-kB binding sites upstream of a minimal E1B promoter-luciferase gene (Stratagene, La Jolla, CA). Transfection efficiency was monitored using a *Renilla* luciferase reporter under control of the thymidine kinase promoter (Promega, Madison, WI). Briefly, $0.5 \times 10^6$ cells were seeded in six-well

**The paper explained**

**Problem**

Complex gene and environment interactions that take place early in life can influence lung development and thus shape how the lung responds to infections later in life. One of the most significant environmental changes the lung will experience takes place at birth when it is first exposed to an oxygen-rich environment. However, too much oxygen (hyperoxia), such as given to preterm infants, can alter postnatal lung development and thus increase the severity of respiratory infections through poorly understood mechanisms. Understanding how hyperoxia at birth increases the severity of respiratory infections is important because it could improve the health of people born preterm and may explain how other gene-environment interactions shape respiratory health.

**Results**

In this study, we show how hyperoxia at birth increases the number of alveolar type 1 (AT1) epithelial cells expressing the proliferation protein Ki-67 in adult mice. Although these cells were not proliferating, they expressed high levels of pro-inflammatory chemokines during infection that drive neutrophils into the lung. While neutrophils play a critical role in defending the lung against infection, too many neutrophils can cause acute lung injury and mortality. Depleting neutrophils or removing Ki-67 from the lung protects the mice from infection, while overexpressing Ki-67 in the lung increases neutrophil recruitment and the severity of respiratory illness. Experiments using a novel mouse AT1-like cell line revealed that Ki-67 enhances pro-inflammatory cytokine signaling, controlling transcription of chemokines that stimulate neutrophils to enter the lung.

**Impact**

Our findings reveal a previously unidentified role for Ki-67 to regulate the intensity of inflammatory gene expression, controlling neutrophil recruitment into the lung. Since too many neutrophils cause acute lung injury and mortality, the severity of respiratory illness can be driven by environmental factors, such as hyperoxia, that increase the expression of Ki-67. Therapies that control the expression of Ki-67 may someday prove beneficial for preventing or alleviating respiratory infectious disease in susceptible individuals.

35-mm dishes and allowed to grow overnight. The cells were then transfected with siRNA to Ki-67 or scrambled control oligos as described above. The cells were then transfected 24 h later with 5 μg of NF-κB-luciferase reporter plasmid and 0.125 μg of *Renilla* luciferase reporter mixed with 50 μg/ml DEAE-dextran in serum-free media. The mixture was added to cells that were subconfluent. At this time, cells were also cultured/in the presence or absence of 0.5 ng/ml recombinant mouse IL-1β/IL-1F2 (401-ML-010/CF from R&D Systems, Minneapolis, MN) for 24 h. Cell extracts were assayed for Firefly and *Renilla* luciferase activity using the Promega Biotech Dual Luciferase Reporter Assay System and a luminometer. NF-κB transcriptional activity was defined as the ratio of Firefly to *Renilla* luciferase activity.

### Graphics

BioRender.com was used to create the graphic Synopsis figure.

### Statistical analysis

No blinding or randomization was done to design the experiments. A minimum of three and a maximum of ten mice were analyzed for each outcome. Data were evaluated using JMP 16 software (SAS Institute, Cary, NC) and graphed as mean ± standard deviation. Graphs showing changes in weight following infection are graphed as mean ± standard error of the mean. A Student's *t* test and a one-way ANOVA followed by Tukey-Kramer HSD test were used to determine overall significance. A Log-Rank (Mantel–Cox) test was used to evaluate differences in mouse survival following infection. The number of samples per analysis shown in the figures reflects biological replicates with $P < 0.05$ considered significant.

## Data availability

Source data for images can be found at https://www.ebi.ac.uk/biostudies/studies/S-BSST1960.

The source data of this paper are collected in the following database record: biostudies:S-SCDT-10_1038-S44321-025-00261-z.

## Peer review information

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

## Acknowledgements

The authors thank Brigid Hogan for sharing *Sftpc^EGFP* transgenic mice (created by John K. Heath, University of Birmingham, UK) and Zea Borok for sharing the *Aqp^Cre* transgenic mice with us. This study was funded in part by US National Institutes of Health grants R01HL091968 (MAO), U01HL122700/U01HL148861 (GSP, RM), R01ES030300 (BPL), R01HL148825 (DAD), R01HL148695 (AR), and a pilot project grant (MAO) from the Environmental Health Science Center (P30ES001247). This Center grant P30ES001247 (BPL) also supported the animal inhalation facility. The French National Cancer Institute (Grant # PLBIO18-094) and the Foundation ARC against cancer (Grant # ARCPGA12021010002850) supported DF.

## Author contributions

**Min Yee**: Conceptualization; Data curation; Formal analysis; Validation; Investigation; Methodology; Writing—review and editing. **Ravi Misra**: Data curation; Formal analysis; Investigation; Methodology. **Sarah Vesecky**: Investigation; Writing—review and editing. **Michael Barravecchia**: Data curation; Formal analysis; Investigation. **Rauf A Najar**: Data curation; Formal analysis; Investigation. **Arshad Rahman**: Formal analysis; Supervision; Investigation; Methodology; Writing—review and editing. **Gloria S Pryhuber**: Supervision; Funding acquisition; Investigation; Writing—review and editing. **David A Dean**: Conceptualization; Formal analysis; Funding acquisition; Writing—review and editing. **B Paige Lawrence**: Conceptualization; Resources; Funding acquisition; Investigation; Writing—review and editing. **Daniel Fisher**: Conceptualization; Resources; Supervision; Funding acquisition; Methodology; Writing—review and editing. **Michael A O'Reilly**: Conceptualization; Data curation; Supervision; Funding acquisition; Methodology; Writing—original draft; Project administration; Writing—review and editing.

Source data underlying figure panels in this paper may have individual authorship assigned. Where available, figure panel/source data authorship is listed in the following database record: biostudies:S-SCDT-10_1038-S44321-025-00261-z.

## Disclosure and competing interests statement

The authors declare no competing interests.

# Expanded View Figures

**Figure EV1.   Identification of alveolar cells expressing Ki-67.**

(**A**) Lungs of adult mice exposed to room air or hyperoxia as neonates were disassociated and stained for Ki-67 and VE-cadherin (CD144), PECAM (CD31) or CD45. (**A**) FACS plots show hyperoxia does not increase Ki-67 in CD144$^+$ or CD31$^+$ endothelial cells. $n = 5$ mice per plot. (**B**) FACS plots showing Ki-67 was not detected in CD45+ leukocytes. $n = 5$ mice per plot. (**C**) Lungs of adult *Sftpc$^{EGFP}$* mice exposed to room air or hyperoxia as neonates were immunostained for Ki-67 (red), EGFP (green), and counterstained with DAPI (blue). Arrow points to rare EGFP$^+$ AT2 cell expressing Ki-67. The proportion of EGFP$^+$ AT2 cells that also express Ki-67 were quantified and graphs as mean ± standard deviation. $n = 10$ mice per group. (Room air vs Hyperoxia: Not significant (NS)). (**D**) Lungs of adult Sftpc$^{EGFP}$ mice exposed to room air or hyperoxia were stained for phospho-histone H3 (Ser10) (red), EGFP (green), and counterstained with DAPI (blue). Arrow points to a phospho-histone H3 (Ser10)$^+$ cell that was rarely detected in mice exposed to room air or hyperoxia. Note that the phospho-histone H3 (Ser10) staining could not be done on the same tissues stained for Ki-67 because both antibodies were made in the same species. The proportion of pHH3+ to EGFP+ cells were quantified and graphed as mean ± standard deviation. $n = 4$ mice per group. (Room air vs Hyperoxia: NS= not significant). Scale bar in (**C**, **D**) = 50 μm. Data reflects biological replicates analyzed by Student's *t* test (**C**, **D**).

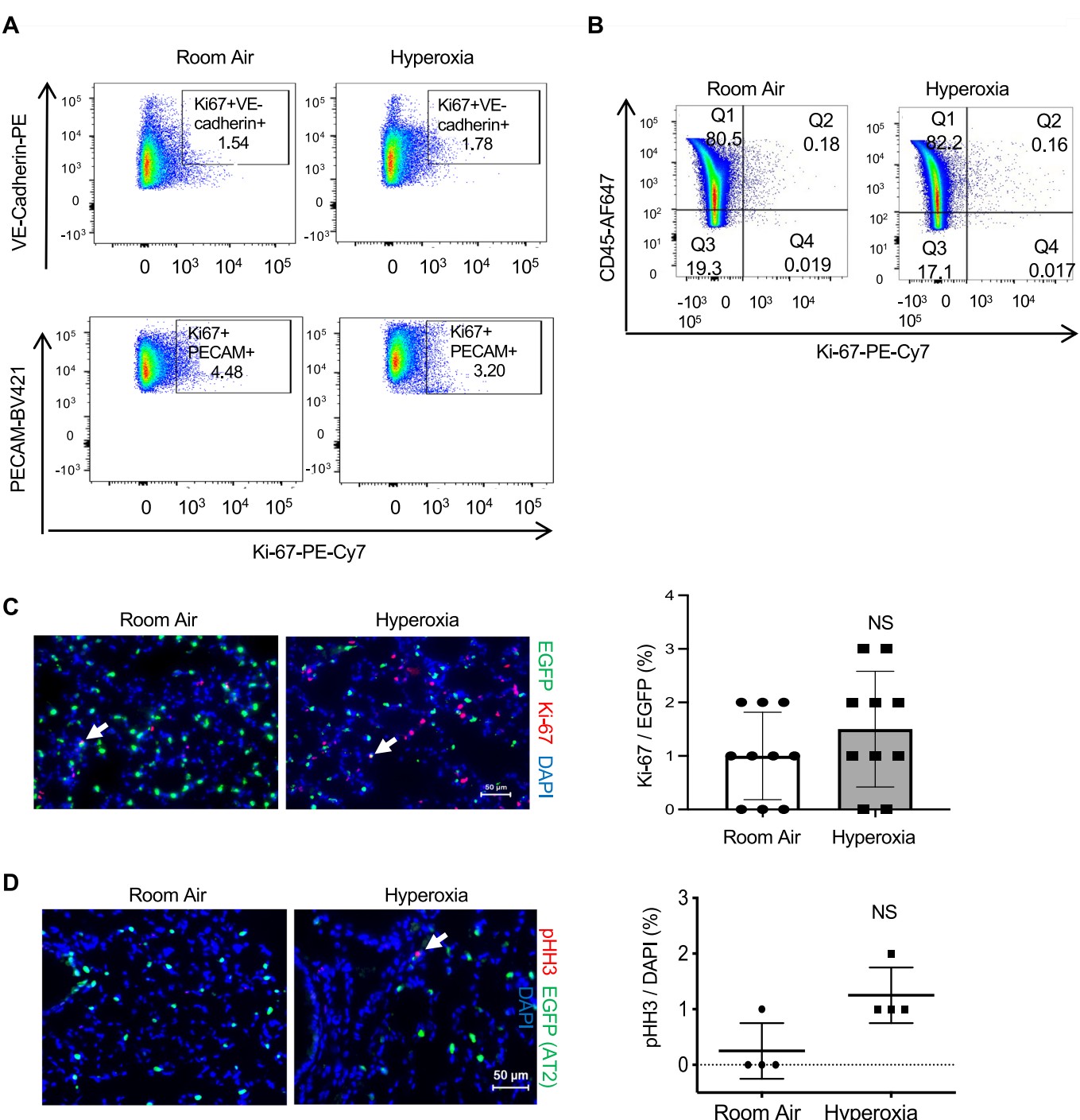

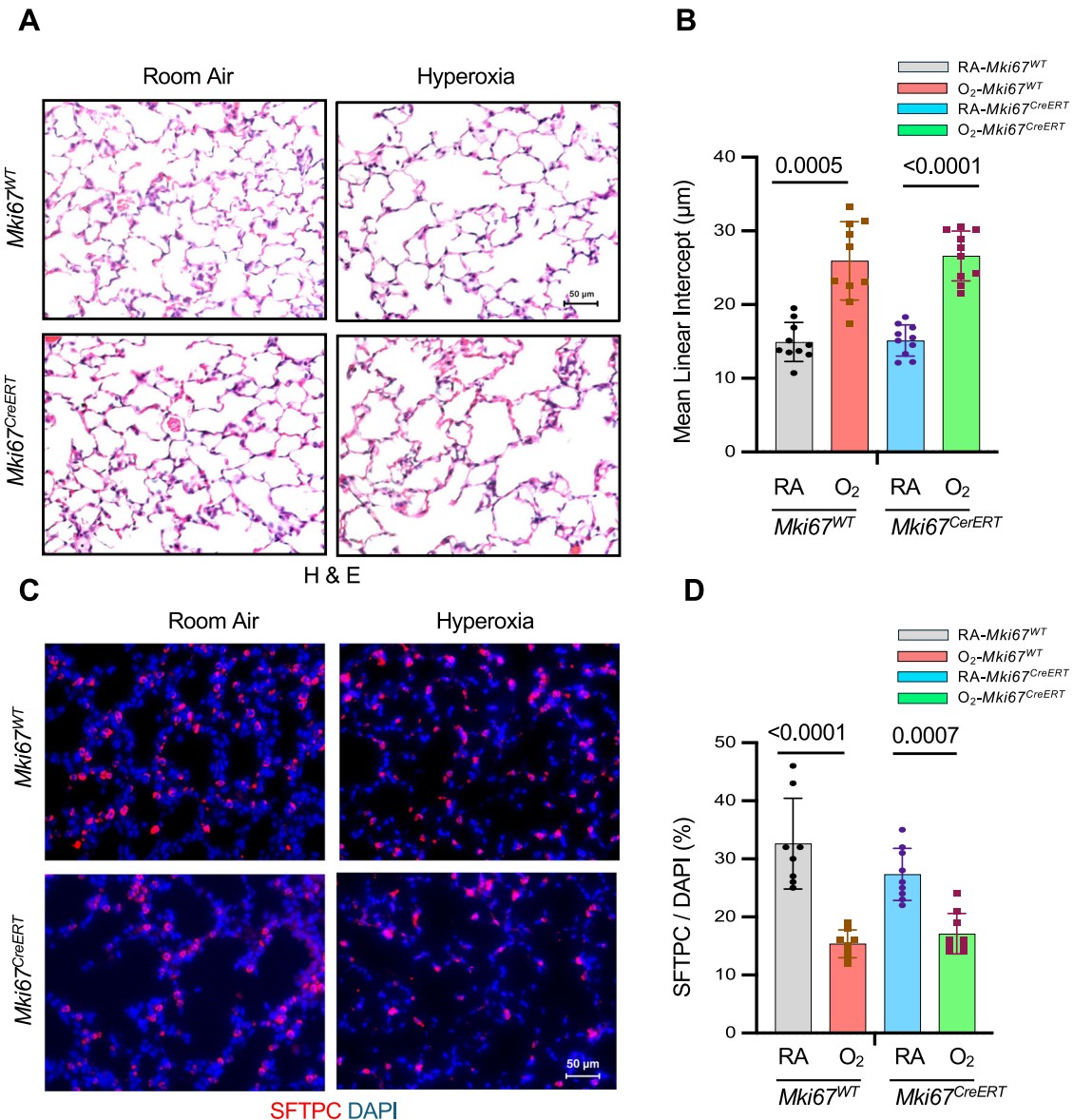

**Figure EV2. Ki-67 is not required for neonatal hyperoxia to disrupt alveolar development.**

(A) H&E stains of lungs from adult *Mki67$^{WT}$* and *Mki67$^{CreERT}$* hypomorph mice exposed to room air or hyperoxia. (B) Mean linear intercept of alveolar size in μm was measured in lungs of adult *Mki67$^{WT}$* and *Mki67$^{CreERT}$* hypomorph mice exposed to room air or hyperoxia and graphed as mean ± standard deviation. $n = 10$ mice per group. (RA-*Mki76$^{WT}$* vs O$_2$ *Mki67$^{WT}$*: $P = 0.0005$; RA *Mki67$^{CreERT}$* vs O$_2$-*Mki67$^{CreERT}$*; $P < 0.0001$). (C) Lungs from adult *Mki67$^{WT}$* and *Mki67$^{CreERT}$* hypomorph mice exposed to room air or hyperoxia were stained for SFTPC (red) and counterstained with DAPI (blue). (D) The proportion of SP-C$^+$ to DAPI$^+$ alveolar cells were quantified and graphed as mean ± standard deviation. $n = 9$ mice per group or 8 for *Mki67$^{WT}$* mice exposed to room air.(RA *Mki67$^{WT}$* vs O$_2$ *Mki67$^{WT}$*: $P < 0.0001$; RA *Mki67$^{CreERT}$* vs O$_2$-*Mki67$^{CreERT}$*; $P = 0.0007$). Data in (B, D) are graphed as mean ± SD with individual samples shown as circles or squares. Scale bar in (A, C) = 50 μm. Data reflect biological replicates analyzed by one-way ANOVA using Tukey-Kramer HSD (B, D).

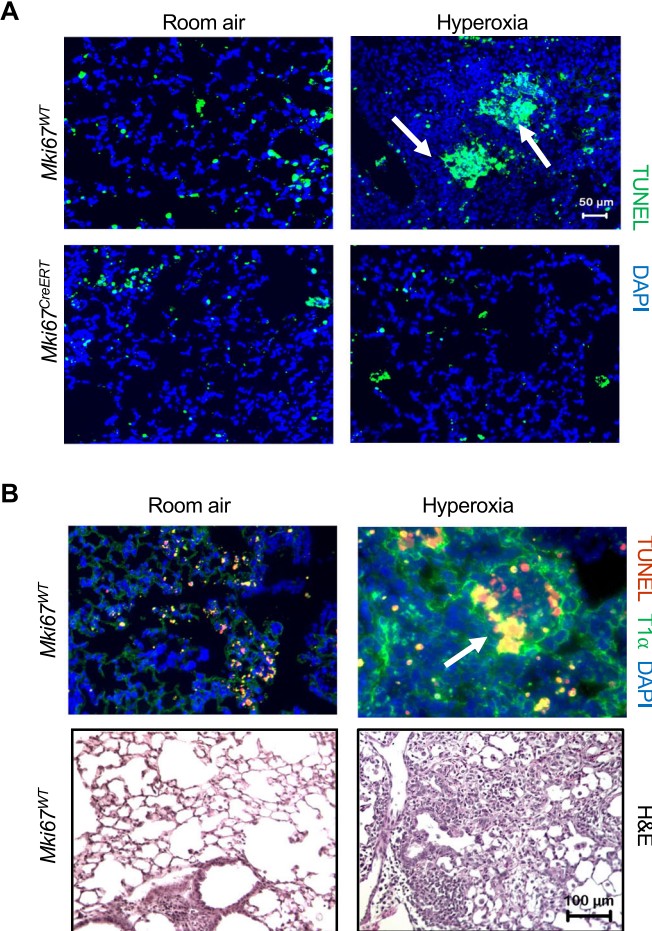

**Figure EV3. Neonatal hyperoxia enhances AT1 apoptosis and NETosis in lungs of infected *Mki67^WT* but not *Mki67^CreERT* hypomorphs.**

Adult *Mki67^WT* and *Mki67^CreERT* mice exposed to room air or hyperoxia were infected with Hkx31 IAV. (A) Lung sections collected on post-infection day 5 were stained with a TUNEL assay (green) used to detect DNA strand breaks and counterstained with DAPI (blue). (B) Adult *Mki67^WT* exposed to room air or hyperoxia were infected with Hkx31 IAV. Lung sections collected on post-infection day 5 were stained for TUNEL (red), antibody to T1α (green) used to detect AT1 cells, and counterstained with DAPI. Arrows point to patches of TUNEL+ cells with pseudo-yellow color in (B) reflecting TUNEL and T1α double-positive cells indicative of AT1 cell death. Images are representative of 10 mice per group with similar pathology. Scale bar in (A) = 100 μm, and (B) = 50 μm.

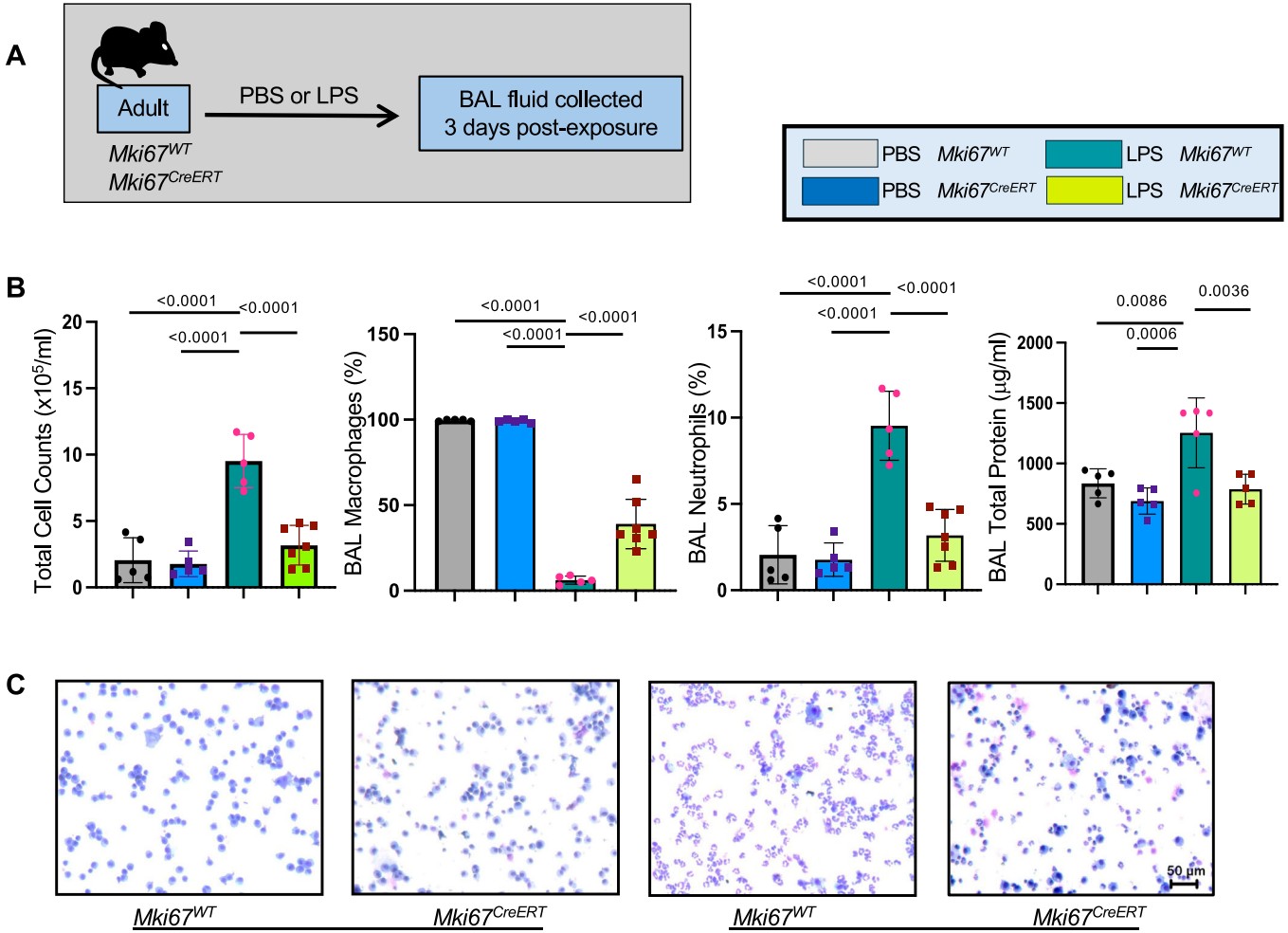

**Figure EV4.  Lung neutrophil recruitment and injury is attenuated in *Mki67^CreERT* hypomorphs exposed to LPS.**

(A) Bronchoalveolar lavages were performed 3 days after LPS (5 mg/kg) or PBS was instilled intratracheally into lungs of adult *Mki67^WT* and *Mki67^CreERT* hypomorph mice. (B) The total number of leukocytes, the proportion of macrophages, the proportion of neutrophils, and total protein were quantified in BAL fluid, and graphed as mean ± standard deviation with individual mice shown in circles or squares. *n* = 5 mice for all groups except 7 *Mki67^CreERT* exposed to LPS. (Total Cell Counts: PBS-*Mki67^WT* vs LPS-*Mki67^WT*: *P* < 0.0001; PBS-*Mki67^CreERT* vs LPS-*Mki67^WT*: *P* < 0.0001; LPS-*Mki67^WT* vs LPS-*Mki67^CreERT*: *P* < 0.0001). (Percent macrophages: PBS-*Mki67^WT* vs LPS-*Mki67^WT*: *P* < 0.0001; PBS-*Mki67^CreERT* vs LPS-*Mki67^WT*: *P* < 0.0001; LPS-*Mki67^WT* vs LPS-*Mki67^CreERT*: *P* < 0.0001). (Percent neutrophils: PBS-*Mki67^WT* vs LPS-*Mki67^WT*: *P* < 0.0001; PBS-*Mki67^CreERT* vs LPS-*Mki67^WT*: *P* < 0.0001; LPS-*Mki67^WT* vs LPS-*Mki67^CreERT*: *P* < 0.0001). (Total Protein: PBS-*Mki67^WT* vs LPS-*Mki67^WT*: *P* = 0.0086; PBS-*Mki67^CreERT* vs LPS-*Mki67^WT*: *P* = 0.0006; LPS-*Mki67^WT* vs LPS-*Mki67^CreERT*: *P* = 0.0036). (C) Representative images of BAL cytospins obtained from the mice. Data in (B) graphed as mean ± standard deviation. Scale bar in (C) = 50 µm. Data reflects biological replicates analyzed by one-way ANOVA using Tukey-Kramer HSD (B).

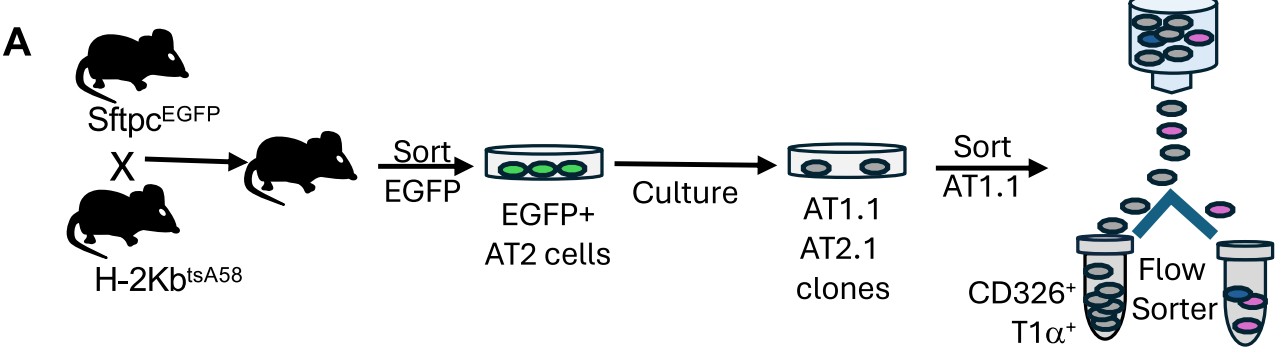

**A**

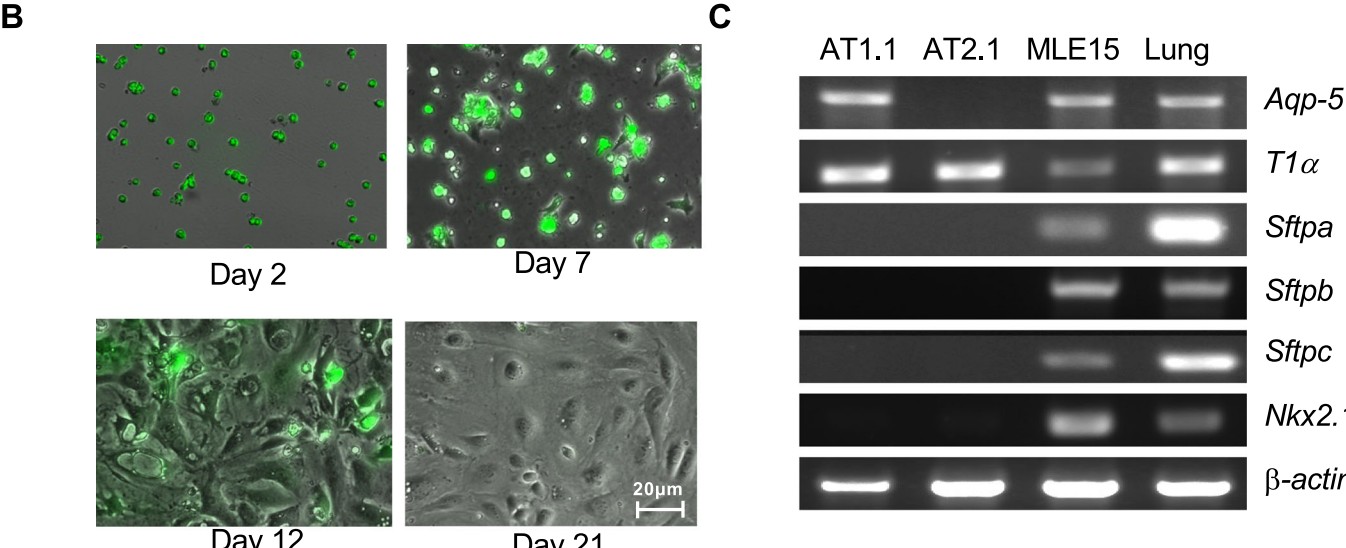

**B**

**C**

**D**

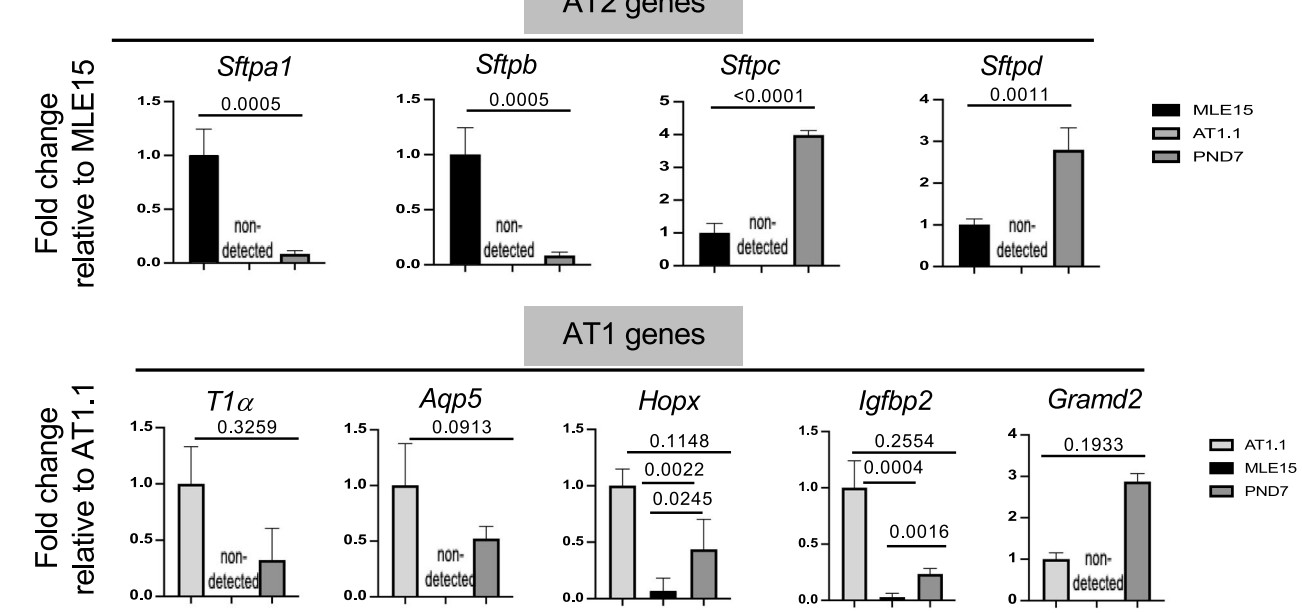

**Figure EV5.  Creation and characterization of mouse AT1.1 cell line.**

(A) Cartoon model showing how mouse AT1.1 cells were created from EGFP + AT2 cells isolated from Immortomice. (B) Images showing loss of green fluorescence as EGFP+ AT2 cells from Immortomice were cultured. (C) PCR was used to detect *Aqp5, T1a, Sftpa, Sftpb, Sftpc, Nkx2.1* and *β-actin* in AT1.1, AT2.1, MLE15 and adult lungs. The amplified products were then visualized by gel electrophoresis. (D) QRT-PCR was used to detect AT1 and AT2-specific genes in AT1.1 cells, MLE15 cells, and postnatal day 7 (PND7) mouse lungs. Data for genes expressed by AT2 cells is graphed as fold change relative to MLE15 cells ± standard deviation. $n = 3$ per condition. (*Sftpa1*: MLE15 vs PND7: $P = 0.0005$; *Sftpb*: MLE15 vs PND7: $P = 0.0005$; *Sftpc*: MLE15 vs PND7: $P < 0.0001$; *Sftpd*: MLE15 vs PND7: $P = 0.0011$). Data for genes expressed by AT1 cells is graphed as fold change relative to AT1.1 cells ± standard deviation. (*T1a*: AT1.1 vs PND7: $P = 0.3259$; *Aqp5*: AT1.1 vs PND7: $P = 0.0913$; *Hopx*: AT1.1 vs MLE15: $P = 0.0022$; AT1.1 vs PND7: $P = 0.1148$; MLE15 vs PND7: $P = 0.0245$; *Ifgbp2*: AT1.1 vs MLE15: $P = 0.0004$; AT1.1 vs PND7: $P = 0.2554$; MLE15 vs PND7: $P = 0.0016$; *Gramd2*: At1.1 vs PND7: $P = 0.1933$). Data reflect biological replicates analyzed by one-way ANOVA using Tukey-Kramer HSD (D).

