## [Peer Review File · EMBO Molecular Medicine]

Ki-67 promotes inflammatory signaling governing neutrophil recruitment during respiratory infections

Min Yee, Ravi Misra, Sarah Vesecky, Michael Barravecchia, Rauf Najar, Arshad Rahman, Gloria Pryhuber, David Dean, B Paige Lawrence, Daniel Fisher, and Michael O'Reilly

Corresponding author: Michael O'Reilly (michael_oreilly@urmc.rochester.edu)

Review Timeline:

Submission Date:	14th Nov 24
Editorial Decision:	10th Jan 25
Revision Received:	13th Apr 25
Editorial Decision:	15th May 25
Revision Received:	28th May 25
Accepted:	30th May 25

Editor: Zeljko Durdevic

Transaction Report:

10th Jan 2025

Dear Prof. O'Reilly,

Thank you for the submission of your manuscript to EMBO Molecular Medicine, and please accept my apologies for the unusual delay in getting back to you. We have now received feedback from two of the three reviewers who agreed to evaluate your manuscript. As the referee #3 will unfortunately not be able to return his/her report in a timely manner, and given that both reviewers provide very similar recommendations, we prefer to make a decision now in order to avoid further delay in the process.

As you will see from their reports pasted below, both referees recognize potential interest of the manuscript but also raise important concerns that should be addressed in a major revision. If you would like to discuss further the points raised by the referees, I am available to do so via email or video. Let me know if you are interested in this option.

We would welcome the submission of a revised version within three months for further consideration. Please let us know if you require longer to complete the revision.

I look forward to receiving your revised manuscript.

Yours sincerely,

Zeljko Durdevic

We require:

- 1) A .docx formatted version of the manuscript text (including legends for main figures, EV figures and tables). Please make sure that the changes are highlighted to be clearly visible.
- 2) Individual production quality figure files as .eps, .tif, .jpg (one file per figure). For guidance, download the 'Figure Guide PDF': (<https://www.embopress.org/page/journal/17574684/authorguide#figureformat>).
- 3) A .docx formatted letter INCLUDING the reviewers' reports and your detailed point-by-point responses to their comments. As part of the EMBO Press transparent editorial process, the point-by-point response is part of the Review Process File (RPF), which will be published alongside your paper.
- 4) A complete author checklist, which you can download from our author guidelines (<https://www.embopress.org/page/journal/17574684/authorguide#submissionofrevisions>). Please insert information in the checklist that is also reflected in the manuscript. The completed author checklist will also be part of the RPF.

6) It is mandatory to include a 'Data Availability' section after the Materials and Methods. Before submitting your revision, primary datasets produced in this study need to be deposited in an appropriate public database, and the accession numbers and database listed under 'Data Availability'. Please remember to provide a reviewer password if the datasets are not yet public (see <https://www.embopress.org/page/journal/17574684/authorguide#dataavailability>).

.

12) Author contributions: You will be asked to provide CRediT (Contributor Role Taxonomy) terms in the submission system. These replace a narrative author contribution section in the manuscript.

13) A Conflict of Interest statement should be provided in the main text.

14) Every published paper now includes a 'Synopsis' to further enhance discoverability. Synopses are displayed on the journal webpage and are freely accessible to all readers. They include a short stand first (maximum of 300 characters, including space) as well as 2-5 one-sentences bullet points that summarizes the paper. Please write the bullet points to summarize the key NEW findings. They should be designed to be complementary to the abstract - i.e. not repeat the same text. We encourage inclusion of key acronyms and quantitative information (maximum of 30 words / bullet point). Please use the passive voice. Please attach these in a separate file or send them by email, we will incorporate them accordingly.

15) Include a Reagents and Tools Table as part of the Methods section, which can be downloaded from our author guidelines (<https://www.embopress.org/page/journal/17574684/authorguide#structuredmethods>)

***** Reviewer's comments *****

Referee #1 (Comments on Novelty/Model System for Author):

The experiments were well designed and comprehensive for the study. Many tools were used to confirm the data obtained. This new role for Ki67 is intriguing but is confusing in view of the existing literature. Additionally, not clear how one might differentiate whether Ki67 is serving as a valid model of proliferation or a modulator of inflammation. Without better understanding how this knowledge could be exploited in diseases, it has limited medical impact. Despite the model system being adequate, whether this is applicable clinically is not clear.

Referee #1 (Remarks for Author):

This paper explores novel roles for Ki67 beyond serving as a maker of proliferation. The authors identify that hypomorphs of Ki67 have decreased inflammatory signaling and neutrophils recruitment compared to WT. This is shown using many approaches.

Overall, the manuscript is sound and provides novel insights, nevertheless, there are contradictions with existing data regarding expression of Ki67 in the lung. In this model, there is no expression of Ki67 beyond the trachea which is confusing yet they evaluate the function of this protein in lung epithelial cells.

Additionally, if Ki67 has many other functions beyond indicating cell proliferation, how can one discern between the forms of ki67 to explain the tight correlation between cell proliferation and expression of Ki67 that others have shown?

Is it also possible that there is a bimodal effect of Ki67? The hypomorphs still have some expression of the gene and protein. Would a low dose have a stabilizing effect on inflammatory signaling and neutrophil recruitment whether a higher dose may serve a different purposes? there are no dose response data here, making difficult to answer. Nevertheless, the use of the various mouse lines is helpful in understanding how these impact Ki67 expression.

The authors need to explain the controversies better and consider a dose response of Ki67 to better understand whether this function is found similarly at low and high doses

Overall, this paper presents novel data but also raises questions around the specificity of this finding and the conflict with years of information regarding the use of Ki67 as a marker of cell proliferation.

Referee #2 (Comments on Novelty/Model System for Author):

The manuscript is well written and the experimental methods are well established. Data quality good. Detailed report given below.

Referee #2 (Remarks for Author):

"Ki-67 promotes inflammatory signaling governing neutrophil recruitment during respiratory infections" is a very interesting manuscript submitted by Yee et al. The manuscript delves into the potential role played by Ki-67 in promoting lung inflammation. The research team has used their well-established model of exposure to hyperoxia as a newborn followed by a second insult later in adulthood. The study has immense strengths as it is modeled on a well-established clinical phenomenon of exaggerated infection in children with BPD. The data presented in the manuscript are quite intriguing and clearly establishes a role for Ki-67 beyond its conventional role of that of a proliferation marker. The manuscript comes at a time when the new putative role of Ki-

67 is coming to light, especially in the field of cancer medicine. The manuscript presents numerous novel data and it's well written, however there are some concerns which may be addressed.

1. Introduction, page 3: "Neutrophils serve as the first line of innate immune defense because they have a broad arsenal of anti-pathogen functions". Concern: Macrophages are also seen to play a significant role in promotion of lung injury especially in second hit pathology. Were any macrophages seen in this model and did Ki67 have any impact on macrophage infiltration?

2. Introduction, page 4: "While mapping how neonatal hyperoxia depletes AT2 cells in the adult lung, we made the unexpected and surprising finding that it increases the number of AT1 cells expressing Ki-67."

Concern: In BPD the surface area of alveoli is less. Does this mean that the total number of AT1 cells are less and the proportion of AT1 with Ki-67 expression has gone up?

3. Results, page 6: "Ki-67 was detected in approximately 1-2% of EGFP+ AT2 cells in adult mice exposed to room air with similar numbers seen in mice exposed to hyperoxia as neonates".

Concern: It is surprising to note that hyperoxia does not increase the proportion/number of Ki67 staining of AT2 cells. It has been shown that hyperoxia triggers proliferation of AT2 cells in the newborn. Is there any explanation for this?

4. Results, page 8: "Thus, neonatal hyperoxia increased the number of AT1 cells expressing Ki-67 from AT2 cells that may have undergone proliferation during hyperoxia but are no longer proliferating".

Concern: AT1 pneumocytes may originate from AT2 or there can be common precursors as well. This aspect may be discussed. (PMID: 36414616, PMID: 37843535)

5. Results, page 8: "Tamoxifen was administered to adult Mki67CreERT; Rosa26mTmG mice that were exposed to room air or hyperoxia has neonates"

Minor concern: Exposed to hyperoxia as neonates. Please correct the 'has'.

6. Results, page 11: "They may also reflect a direct involvement of Ki-67 in modulating inflammatory responses in AT1 cells because increased CXCL5 staining overlapped with T1 alpha expressed by AT1 cells"

Concern: There is an association of expression of Ki67 with the concentration of mRNA of mediators of inflammation. Direct involvement is uncertain or there seems to be insufficient data to prove that.

7. Results, page 12: "Neonatal hyperoxia therefore increases the severity of IAV infections by stimulating neutrophil recruitment in a Ki-67-dependent manner".

Concern: Anti Lys6G seems to be down stream of Ki67 and hence irrespective of the ki67 status may suppress neutrophil infiltration. This conclusion seems to be an overinterpretation of the data.

8. Discussion, page 15: "neonatal AT2 cells do not produce adult AT1 cells during postnatal growth and development (Penkala et al, 2021; Yee et al, 2016)".

Concern: This sentence seems to contradict the fact that AT2 give rise to AT1 cells.

9. Discussion, page 17: "AT1 and AT2 cells expressing Ki-67 might be preferentially killed off during infection because they express higher levels of Cxcl genes that attract neutrophils. Hypothetically loss of these cells would then extinguish high expression of Cxcl genes in the alveolar space, thus dampening or shutting off neutrophil recruitment."

Concern: This speculation does not have enough evidence. Kindly explain.

EMBO Molecular Medicine: EMM-2024-20920

Title: Ki-67 promotes inflammatory signaling governing neutrophil recruitment during respiratory infections

Authors: Yee, et al.

Reviewer 1

We appreciate that Reviewer 1 felt that our experiments were “well designed, comprehensive, and provided a new intriguing role for Ki-67”. However, they also felt the findings “did not distinguish a role for Ki-67 in proliferation versus as a modulator of inflammation”. This confusion was felt to limit clinical applicability. We apologize for not clarifying the role of Ki-67 in proliferation and inflammation. Ki-67 was identified as a protein expressed by proliferating cells. However, as more clearly explained in our revised manuscript, Ki-67 has been detected in non-proliferating cells, silencing expression rarely impacts proliferation, and it has been recently established that Ki-67 does not control cell proliferation. Our finding that Ki-67 promotes pro-inflammatory gene expression during respiratory infections supports a growing body of evidence showing Ki-67 interacts with chromatin and modulates the intensity of gene expression in a cell-specific manner. Our finding that Ki-67 enhances inflammatory signaling in the quiescent adult mouse lung suggests that its ability to shape inflammation is not restricted to proliferating cells. The clinical implications of these findings are expanded upon in the revised discussion.

Concern 1. In this model, there is no expression of Ki67 beyond the trachea which is confusing, yet they evaluate the function of this protein in lung epithelial cells.

Response 1. Thank you for this comment. Our paper focuses on Ki-67 expression in distal alveolar type (AT) 1 and 2 epithelial cells because this is where it was detected histologically and by flow cytometry. Ki-67 was not detected in mainstem bronchi, proximal or distal bronchiolar (airway) cells. We did not look for Ki-67 in the trachea because hyperoxia is a freely diffusible gas that primarily toxifies distal alveolar but not tracheal or proximal airway cells.

Concern 2. Additionally, if Ki67 has many other functions beyond indicating cell proliferation, how can one discern between the forms of ki67 to explain the tight correlation between cell proliferation and expression of Ki67 that others have shown?

Response 2. We apologize if our statements about Ki-67's role in proliferation versus inflammation were confusing. Ki-67 was originally identified by a monoclonal antibody generated in mice injected with non-Hodgkin lymphoma cell line L438. While Ki-67 is detected in proliferating cells, it has also been detected in post-mitotic and quiescent cells, and silencing Ki-67 expression rarely affects cell proliferation (as also shown by the fact that the Ki-67 mutant mice we used are viable). Our study shows that alveolar epithelial cells expressing Ki-67 are not proliferating, as defined by the lack of BrdU or phospho-histone H3 staining. Instead, Ki-67 enhances pro-inflammatory signaling driving neutrophil recruitment during respiratory infections. We believe this function is independent of cell proliferation because 1) Ki-67 is not required for cell proliferation, 2) Ki-67 was detected in adult AT1 cells that are not proliferating, and 3) silencing Ki-67 expression in AT1.1 cells affects IL-1 β signaling without disrupting cell proliferation. However, it is theoretically possible that Ki-67 modulates inflammation differently when expressed by a proliferating cell. We reflect on this possibility in the discussion.

Concern 3. Is it also possible that there is a bimodal effect of Ki67? The hypomorphs still have some expression of the gene and protein. Would a low dose have a stabilizing effect on inflammatory

signaling and neutrophil recruitment while a higher dose may serve a different purpose? there are no dose response data here, making difficult to answer. Nevertheless, the use of the various mouse lines is helpful in understanding how these impact Ki67 expression.

Response 3. The reviewer's suggestion that different levels of Ki-67 within the cell exert unique biological functions is an interesting concept. We have no evidence that the intensity of Ki-67 staining or the intracellular localization between proliferating and non-proliferating cells varies. In other words, the intensity of Ki-67 staining seen in the quiescent adult lung approximates the intensity of staining seen in proliferating cells detected during postnatal lung development. We believe it is beyond the scope of the current study to investigate whether and how different levels of Ki-67 within the cell influence its function. This would require generation of a Ki-67 null cell line or mouse in which Ki-67 expression is restored in a tightly controlled manner.

Concern 4. The authors need to explain the controversies better and consider a dose response of Ki67 to better understand whether this function is found similarly at low and high doses.

Response 4. Thank you for this excellent suggestion. Figure 3 shows that the number of Ki-67+ cells detected in the lung is extremely low in *Mki67^{CreERT}* (hypomorph) mice and progressively increases in *Mki67^{WT/CreERT}* (heterozygous for Ki-67 gene) and *Mki67^{WT}* (homozygous wildtype for Ki-67 gene). This allowed us to test whether the severity of acute lung injury is modulated by the dose (number) of cells expressing Ki-67. New data in Figure 9 shows that infected *Mki67^{WT/CreERT}* (heterozygote) mice have an intermediate response between the *Mki67^{WT}* (wildtype) and *Mki67^{CreERT}* (hypomorphs) as defined by survival, weight loss, neutrophil recruitment, and pathologic lung disease. Thus, neutrophil recruitment and the severity of acute lung injury following PR8 infection correlates with increasing number of Ki-67+ alveolar cells. Neonatal hyperoxia increases the number of Ki-67+ cells above that normally seen in mice exposed to room air and thus increases neutrophil recruitment when mice are infected with Hkx31, a homologous strain of PR8 that poorly recruits neutrophils in control mice. Our findings suggest that higher "doses" of Ki-67 (defined as number of cells expressing Ki-67) increases the severity of respiratory infections.

Reviewer 2

Reviewer 2 felt our manuscript was "well-written, the study had many strengths, and the quality of data was good". However, they were concerned that our study failed to consider AT1 to AT2 differentiation and pointed out statements that were over-interpreted or unclear. We thank the reviewer for their kind remarks about our manuscript. We agree that some investigators have shown AT1 cells may produce AT2 cells. However, this is a controversial conclusion because, to our knowledge and personal discussion with Dr. Zea Borok, all AT1-lineage drivers target a small population of AT2 and airway ciliated cells. As example, the *Aqp5^{Cre}* mouse used to map Ki-67 in AT1 cells also targets a small number of AT2 cells (PMCID: PMC2937230). Whether this reflects "leaky expression" or a true lineage relationship is not clear at this time. Our revised manuscript discusses the idea that AT1 progenitors that expand during hyperoxia may also be the source of adult AT2 cells expressing Ki-67.

Concern 1. Introduction, page 3: "Neutrophils serve as the first line of innate immune defense because they have a broad arsenal of anti-pathogen functions". Concern: Macrophages are also seen to play a significant role in promotion of lung injury especially in second hit pathology. Were any macrophages seen in this model and did Ki67 have any impact on macrophage infiltration?

Response 1. This is a very logical question. We previously reported that macrophage numbers are

not different between naïve adult mice exposed to room air or hyperoxia as neonates (PMCID: PMC2383992). They increase 3-5 days after infection with higher numbers persisting in adult mice exposed to hyperoxia as neonates. Despite higher numbers, a former postdoctoral fellow working with Dr. Rusty R. Elliott (a macrophage expert now working at the University of Virginia-Charlottesville) did not detect any change in macrophage phenotype or phagocytic activity. Because mice lacking Ki-67 do not get sick when infected with virus, we are currently unable to determine whether Ki-67 shapes macrophage infiltration at these later time points.

Concern 2. Introduction, page 4: "While mapping how neonatal hyperoxia depletes AT2 cells in the adult lung, we made the unexpected and surprising finding that it increases the number of AT1 cells expressing Ki-67." Concern: In BPD the surface area of alveoli is less. Does this mean that the total number of AT1 cells are less and the proportion of AT1 with Ki-67 expression has gone up?

Response 2. Neonatal hyperoxia increases the number of AT1 cells as defined by increased number of Hopx+ cells (this study) and expression of T1 α /Pdpn1 (immunostaining and western blot in prior studies). We conclude that neonatal hyperoxia increases the number of AT1 cells. We also see increased expression of AT1 cells expressing Ki-67. But we cannot prove that Ki-67 is expressed only by those AT1 cells that were created by hyperoxia, although this is an interesting possibility that we expand upon in the discussion.

Concern 3. Results, page 6: "Ki-67 was detected in approximately 1-2% of EGFP+ AT2 cells in adult mice exposed to room air with similar numbers seen in mice exposed to hyperoxia as neonates". It is surprising to note that hyperoxia does not increase the proportion/number of Ki67 staining of AT2 cells. It has been shown that hyperoxia triggers proliferation of AT2 cells in the newborn. Is there any explanation for this?

Response 3. We previously showed in newborn mice that hyperoxia rapidly stimulates and then inhibits proliferation of AT2 cells (PMID: 16861382 and PMCID: PMC4860080). AT2 cells are then slowly depleted by 50-70% when neonatal mice exposed to hyperoxia return to room air. As shown in Figure 2B of the current manuscript, lineage tracing using *Sftpc*^{CreERT} mice shows that these cells differentiate into AT1 cells. This helps explain why adult mice exposed to hyperoxia as neonates have fewer AT2 and more AT1 cells than control mice exposed to room air. Since hyperoxia does not increase the proportion of Ki-67 stained AT2 cells, we originally thought this just reflected stochastic chance. However, responding to concern 4 reminded us Ki-67 present in AT2 cells might come from an AT1 progenitor. Given the limitations of sorting this out, these two possibilities are presented in the revised discussion.

Concern 4. Results, page 8: "Thus, neonatal hyperoxia increased the number of AT1 cells expressing Ki-67 from AT2 cells that may have undergone proliferation during hyperoxia but are no longer proliferating". AT1 pneumocytes may originate from AT2 or there can be common precursors as well. This aspect may be discussed. (PMID: 36414616, PMID: 37843535)

Response 4. As shown in our previous paper and in this study, *Sftpc*+ AT2 cells that are lineage labeled at birth using *Sftpc*^{CreERT} mice produce adult AT2 cells but rarely any adult AT1 cells. Please note that we are referring to mice exposed only to room air. These findings suggests that a common progenitor of AT1/AT2 cells defined by *Sftpc* does not exist after birth. But as correctly pointed out by the reviewer, a common AT1/AT2 lineage defined by Hopx or another AT1 marker may exist because lineage labeling AT1 cells at birth using *Hopx*^{CreERT} mice will mark a small number of adult AT2 cells. Whether this reflects a true lineage or a leak in gene expression is not clear (personal communication with Zea Borok) and has created controversy in the field. Our studies do not reconcile this controversy because the *Aqp5*^{Cre} mouse used in our study targets a small number of AT2 cells. Our

revised discussion reflects on the possibility that Ki-67 detected in AT2 cells may come from pre-existing AT2 cells or from this common AT1/AT2 progenitor defined by an AT1 marker.

Concern 5. Results, page 8: "Tamoxifen was administered to adult Mki67CreERT; Rosa26mTmG mice that were exposed to room air or hyperoxia has neonates". Minor concern: Exposed to hyperoxia as neonates. Please correct the 'has' (on page 8 of the results)

Response 5. Thank you. We have corrected the grammar.

Concern 6. Results, page 11: "They may also reflect a direct involvement of Ki-67 in modulating inflammatory responses in AT1 cells because increased CXCL5 staining overlapped with T1 alpha expressed by AT1 cells". There is an association of expression of Ki67 with the concentration of mRNA of mediators of inflammation. Direct involvement is uncertain or there seems to be insufficient data to prove that.

Response 6. Thank you. We agree that the word "involvement" is too strong and have deleted it.

Concern 7. Results, page 12: "Neonatal hyperoxia therefore increases the severity of IAV infections by stimulating neutrophil recruitment in a Ki-67-dependent manner". Concern: Anti Lys6G seems to be down stream of Ki67 and hence irrespective of the ki67 status may suppress neutrophil infiltration. This conclusion seems to be an overinterpretation of the data.

Response 7. We agree and have removed the words "in a Ki-67-dependent manner". Anti-Lys6G was used to demonstrate that the excessive influx of neutrophils was contributory to acute lung injury seen when mice are infected with influenza A virus.

Concern 8. Discussion, page 15: "neonatal AT2 cells do not produce adult AT1 cells during postnatal growth and development (Penkala et al, 2021; Yee et al, 2016)". Concern: This sentence seems to contradict the fact that AT2 give rise to AT1 cells.

Response 8. Stating that AT2 cells do not produce AT1 cells does seem like a contradiction. However, as pointed out in Response 4, our prior study (Yee et al., 2016) and the current manuscript show that AT2 cells that are lineage labeled at birth using *Sftpc*^{CreERT} mice will only produce adult AT2 cells. Our findings are consistent with those of Penkala et al, 2021 whose abstract states "the ability of AT2 cells to regenerate AT1 cells is restricted to the adult lung". Why AT2 cells at birth do not produce AT1 cells in the adult mouse remains to be determined. Hypothetically, AT2 cells only produce AT1 cells after alveolar injury, or this ability occurs as the lung matures postnatally.

Concern 9. Discussion, page 17: "AT1 and AT2 cells expressing Ki-67 might be preferentially killed off during infection because they express higher levels of Cxcl genes that attract neutrophils. Hypothetically loss of these cells would then extinguish high expression of Cxcl genes in the alveolar space, thus dampening or shutting off neutrophil recruitment." Concern: This speculation does not have enough evidence. Kindly explain.

Response 9. We agree this was too speculative and have deleted this statement.

15th May 2025

Dear Prof. O'Reilly,

Thank you for the submission of your revised manuscript to EMBO Molecular Medicine. We have now heard back from the one referee who agreed to evaluate your manuscript. This referee also assessed author responses to concerns raised by the referee #2. I am pleased to inform you that we will be able to accept your manuscript pending the following final amendments:

1) Authors: There is name discrepancy for Sarah Veseky in the manuscript vs. Sarah Veseky in our system. Please correct.

2) Figures:

- During a standard image analysis, we detected potential aberrations in the figure set, and we would like to clarify these issues before accepting your manuscript for publication. We kindly invite you to check images in Figure 3C and explain the overlap between Mki67WT/CreERT-Room Air and Mki67CreERT-Hypoxia. Source data indicate that these images are from the same sample but slightly different area. Please clarify and correct and make sure that all figures are accurate.

- Main figures and EV figures should be uploaded as individual high-resolution files. Please rename EV figures to Figure EV1 etc. in the legends, main text and file names. Please check "Author Guidelines" for more information:

<https://www.embopress.org/page/journal/17574684/authorguide#figureformat>

<https://www.embopress.org/page/journal/17574684/authorguide#expandedview>

3) In the main manuscript file, please do the following:

- Please address all comments suggested by our data editors listed below:

o Data availability statement:

1. The specific URL for S-BSST1960 dataset needs to be provided in the data availability section.

o Figure legends:

1. Please note that the exact p values are not provided in the legends of figures 1B, C; 2B, 3C, E; 5A-D; 6A, B, D-F; 7C, E; 8B, 9D, 10C, D; EV2 B, D; EV 4B.

2. Please note that information related to n is missing in the legends of EV2B, D; EV 4B, EV 5D.

3. Please note that the error bars are not defined in the legends of figures 4B, 8D, 9C, D; 10C, D, EV1D, EV 5D.

4. Please note that scale bar and its definition are missing for EV 5B.

- Add up to 5 keywords.

- Add callouts for Figure 5D.

- In Methods, add the following paragraph:

Graphics:

(some of the... OR Figure #... OR synopsis) Graphics were created with BioRender.com.

- Indicate in legends exact n and exact p values, not a range, along with the statistical test used. To keep the figures "clear" some authors found providing an Appendix table Sx with all exact p-values preferable. You are welcome to do this if you want to.

- Please provide a Reagents and Tools Table (should be uploaded as a separate file) followed by a Methods and Protocols section. More information on how to adhere to this format as well as downloadable templates (.docx) for the Reagents and Tools Table can be found in our author guidelines: <https://www.embopress.org/page/journal/17574684/authorguide#structuredmethods>

An example of a paper with Structured Methods can be found here:

<https://www.embopress.org/doi/full/10.1038/s44320-024-00037-6#sec-4>

- Please remove the sentence "Source data for flow cytometry and graphs were provided during submission" from "Data availability" and use the following format to report the accession number of your deposited data:

[data type]: [full name of the resource] [accession number/identifier] ([doi or URL or identifiers.org/DATABASE:ACCESSION])

Please check "Author Guidelines" for more information.

<https://www.embopress.org/page/journal/17574684/authorguide#availabilityofpublishedmaterial>

4) Funding: Please make sure that information about all sources of funding are complete in both our submission system and in the manuscript. Currently, R01ES030300, R01HL148695 and the Foundation ARC against cancer (Grant # ARCPGA12021010002850) are missing in our submission system. Please correct.

5) Synopsis:

- Synopsis image: Please provide the image as a high-resolution jpeg file 550 px-wide x (300-600)-px high. Remove the BioRender reference from the image.

- Synopsis text: Please remove it from the main manuscript and upload it as a separate .doc file.

6) As part of the EMBO Publications transparent editorial process initiative (see our Editorial at

<http://embomolmed.embopress.org/content/2/9/329>), EMBO Molecular Medicine will publish online a Review Process File (RPF) to accompany accepted manuscripts. This file will be published in conjunction with your paper and will include the anonymous referee reports, your point-by-point response and all pertinent correspondence relating to the manuscript. Let us know whether

you agree with the publication of the RPF and as here, if you want to remove or not any figures from it prior to publication. Please note that the Authors checklist will be published at the end of the RPF.

7) Please provide a point-by-point letter INCLUDING my comments as well as the reviewer's reports and your detailed responses (as Word file).

I look forward to reading a new revised version of your manuscript as soon as possible.

Yours sincerely,

Zeljko Durdevic

Zeljko Durdevic
Senior Editor
EMBO Molecular Medicine

*** Instructions to submit your revised manuscript ***

- 1) a .docx formatted version of the manuscript text (including Figure legends and tables)
- 2) Separate figure files*
- 3) supplemental information as Expanded View and/or Appendix. Please carefully check the authors guidelines for formatting Expanded view and Appendix figures and tables at <https://www.embopress.org/page/journal/17574684/authorguide#expandedview>
- 4) a letter INCLUDING the reviewer's reports and your detailed responses to their comments (as Word file).
- 5) The paper explained: EMBO Molecular Medicine articles are accompanied by a summary of the articles to emphasize the major findings in the paper and their medical implications for the non-specialist reader. Please provide a draft summary of your article highlighting
 - the medical issue you are addressing,
 - the results obtained and
 - their clinical impact.This may be edited to ensure that readers understand the significance and context of the research. Please refer to any of our published articles for an example.
- 6) Author contributions: the contribution of every author must be detailed in a separate section.

7) EMBO Molecular Medicine now requires a complete author checklist (<https://www.embopress.org/page/journal/17574684/authorguide>) to be submitted with all revised manuscripts. Please use the checklist as guideline for the sort of information we need WITHIN the manuscript. The checklist should only be filled with page numbers where the information can be found. This is particularly important for animal reporting, antibody dilutions (missing) and exact values and n that should be indicated instead of a range.

8) Every published paper now includes a 'Synopsis' to further enhance discoverability. Synopses are displayed on the journal webpage and are freely accessible to all readers. They include a short stand first (maximum of 300 characters, including space) as well as 2-5 one sentence bullet points that summarise the paper. Please write the bullet points to summarise the key NEW findings. They should be designed to be complementary to the abstract - i.e. not repeat the same text. We encourage inclusion of key acronyms and quantitative information (maximum of 30 words / bullet point). Please use the passive voice. Please attach these in a separate file or send them by email, we will incorporate them accordingly.

You are also welcome to suggest a striking image or visual abstract to illustrate your article. If you do please provide a jpeg file 550 px-wide x 300-600px high.

9) A Conflict of Interest statement should be provided in the main text

10) Please note that we now mandate that all corresponding authors list an ORCID digital identifier. This takes <90 seconds to complete. We encourage all authors to supply an ORCID identifier, which will be linked to their name for unambiguous name identification.

Currently, our records indicate that the ORCID for your account is 0000-0001-5094-696X.

Link Not Available

11) Include a Reagents and Tools Table as part of the Methods section, which can be downloaded from our author guidelines (<https://www.embopress.org/page/journal/17574684/authorguide#structuredmethods>)

Photos 400-800 DPI

*Additional important information regarding figures and illustrations can be found at <https://bit.ly/EMBOPressFigurePreparationGuideline>. See also figure legend preparation guidelines: <https://www.embopress.org/page/journal/17574684/authorguide#figureformat>

***** Reviewer's comments *****

Referee #1

Remarks for Author:

The authors have addressed the queries satisfactorily.

The authors addressed the remaining editorial issues.

30th May 2025

Dear Prof. O'Reilly,

We are pleased to inform you that your manuscript is accepted for publication and is now being sent to our publisher to be included in the next available issue of EMBO Molecular Medicine.

Zeljko Durdevic
Senior Editor
EMBO Molecular Medicine
